# Fluvastatin Converts Human Macrophages into Foam Cells with Increased Inflammatory Response to Inactivated *Mycobacterium tuberculosis H37Ra*

**DOI:** 10.3390/cells13060536

**Published:** 2024-03-18

**Authors:** María Teresa Montero-Vega, Joaquín Matilla, Eulalia Bazán, Diana Reimers, Ana De Andrés-Martín, Rafael Gonzalo-Gobernado, Carlos Correa, Francisco Urbano, Diego Gómez-Coronado

**Affiliations:** 1Servicio de Bioquímica-Investigación, Hospital Universitario Ramón y Cajal, Instituto Ramón y Cajal de Investigación Sanitaria (IRYCIS), 28034 Madrid, Spain; joaquin.matilla@telefonica.net (J.M.); diego.gomez@hrc.es (D.G.-C.); 2Servicio de Neurobiología-Investigación, Hospital Universitario Ramón y Cajal, Instituto Ramón y Cajal de Investigación Sanitaria (IRYCIS), 28034 Madrid, Spain; bazanizqui@gmail.com (E.B.); dianareimers2@gmail.com (D.R.); 3Servicio de Inmunología, Hospital Universitario Ramón y Cajal, Instituto Ramón y Cajal de Investigación Sanitaria (IRYCIS), 28034 Madrid, Spain; aandresm@salud.madrid.org; 4Departamento de Biología Molecular y Celular, Centro Nacional de Biotecnología (CNB), Consejo Superior de Investigaciones Científicas (CSIC), 28049 Madrid, Spain; rd.gonzalo@cnb.csic.es; 5Unidad de Cirugía Experimental y Animalario, Investigación, Hospital Universitario Ramón y Cajal, Instituto Ramón y Cajal de Investigación Sanitaria (IRYCIS), 28034 Madrid, Spain; carlos.correa53@hotmail.com; 6Servicio Interdepartamental de Investigación (SIdI), Facultad de Medicina, Universidad Autónoma, 28029 Madrid, Spain; francisco.urbano@uam.es

**Keywords:** tuberculosis, statins, host-directed therapy, foam cells, granulomas, NLRP3 inflammasome, mevalonate-kinase deficiencies

## Abstract

Cholesterol biosynthesis inhibitors (statins) protect hypercholesterolemic patients against developing active tuberculosis, suggesting that these drugs could help the host to control the pathogen at the initial stages of the disease. This work studies the effect of fluvastatin on the early response of healthy peripheral blood mononuclear cells (PBMCs) to inactivated *Mycobacterium tuberculosis (Mtb) H37Ra*. We found that in fluvastatin-treated PBMCs, most monocytes/macrophages became foamy cells that overproduced NLRP3 inflammasome components in the absence of immune stimulation, evidencing important cholesterol metabolism/immunity connections. When both fluvastatin-treated and untreated PBMCs were exposed to *Mtb H37Ra*, a small subset of macrophages captured large amounts of bacilli and died, concentrating the bacteria in necrotic areas. In fluvastatin-untreated cultures, most of the remaining macrophages became epithelioid cells that isolated these areas of cell death in granulomatous structures that barely produced IFNγ. By contrast, in fluvastatin-treated cultures, foamy macrophages surrounded the accumulated bacteria, degraded them, markedly activated caspase-1 and elicited a potent IFNγ/cytotoxic response. In rabbits immunized with the same bacteria, fluvastatin increased the tuberculin test response. We conclude that statins may enhance macrophage efficacy to control *Mtb*, with the help of adaptive immunity, offering a promising tool in the design of alternative therapies to fight tuberculosis.

## 1. Introduction

The bacillus *Mtb* is the etiological agent of tuberculosis, one of the leading causes of death worldwide throughout history [1]. In response to *Mtb* infection, host immunity forms granulomas, an ancient response that restricts and kills the pathogen with variable efficacy [2]. Different authors have evidenced that virulent *Mtb* strains exploit the early stages of granulomas formation for their proliferation and expansion, also restricting the development of a specific immune response, thus determining the course of the disease [3]. Studies in mice have revealed that, in the incipient granuloma, some foamy macrophages (FMs) recruited to clean the immune debris generated become infected by *Mtb*. The pathogen interferes with lipid metabolism of these infected cells, converting them to lipid-laden FMs (LL-FMs) that are unable to contain the infection [4]. In these macrophages, the bacilli induce the accumulation of cholesterol, cholesteryl ester, and triacylglycerides, and use them as carbon and energy sources, and to upregulate genes associated with production of virulence factors, evasion of immunity and drug resistance. In this way, the pathogen persists latently in the infected macrophages [5,6]. In human granulomas, numerous LL-FMs locate at the interface region nearest to the caseum where *Mtb* concentrates [7]. When these macrophages die, they deliver their lipidic cargo to these necrotic areas, creating a lipid-rich niche where the bacilli survive protected from host immunity [8]. Overtime, these LL-FMs become key players for the persistence of *Mtb* in the granuloma, also contributing to its destabilization and to the dissemination of the bacilli. Thus, LL-MFs represent an important therapeutic target in fighting tuberculosis [9,10].

In the search for pre-existing drugs that could be efficient against *Mtb*, statins, the cholesterol biosynthesis inhibitors prescribed to hypercholesterolemic patients, have shown promising beneficial effects [11,12,13]. Different systematic reviews and meta-analyses, strongly associate the use of statins with a lower risk of developing active tuberculosis [14,15,16], suggesting that these drugs could help the host to control the pathogen at the first stages of the infection. At an experimental level, different studies have shown that statins reduce the severity of infectious diseases caused by *Mtb* and other pathogens not only by reducing cholesterol levels (both in the host and in pathogens), but also by exerting poorly known immunomodulatory actions [17,18,19]. In addition, other authors have hypothesized a potential beneficial effect of statins in the management of the immune reconstitution syndrome [20]. Statins inhibit the activity of 3-hydroxy-3-methyl-glutaryl-coenzyme A (HMG-CoA) reductase, the rate-limiting enzyme of cholesterol biosynthesis. Moreover, through this inhibition, statins also interfere with the synthesis of other isoprenoids, which are essential molecules for different cellular processes [21,22]. Two decades ago, we were pioneers in proposing that statins could be an adjuvant therapy for tuberculosis by promoting caspase-1 activation and the release of interleukin (IL)-1β and IL-18 [23], two essential ILs for *Mtb* control [24]. More recently, different studies have provided evidence that statins prevent the prenylation of some unknown proteins that negatively regulate the assembly of inflammasomes [25], the structures that activate caspase-1 [26]. Inflammasome assembly requires two cell signals. Signal 1 is generated by membrane pattern recognition receptors (PRRs) after sensing extracellular pathogen associated molecular patterns (PAMPs), and/or danger associated molecular patterns (DAMPs), and induces up-regulation of the inflammasome components, pro-caspase-1 and pro-IL-1β. Signal 2 is mediated by PAMPs, DAMPs, cellular stressors and even metabolic perturbations, and induces a conformational rearrangement of a specific cytosolic PRR, thereby initiating its assemblage. Most inflammasomes oligomerize and assemble with ASC (apoptosis-associated speck-like protein containing a CARD), an adaptor protein that recruits pro-caspase-1 to the structure for their autoprocessing and activation [27,28,29]. Some authors have proposed that virulent *Mtb* strains impede the activation of the Nucleotide-binding oligomerization domain and Leucine-rich repeat-containing Receptor Pyrin domain-containing protein -3 (NLRP3) inflammasome, predisposing the host to acquire the disease [30,31,32]. Furthermore, human mutations linked to enhanced NLRP3 inflammasome activity limit *Mtb* growth in the infected macrophages [33]. 

The aim of this work was to study the effect of the drug fluvastatin on the early immune response elicited by peripheral blood mononuclear cells (PBMCs) from healthy donors to inactivated *Mtb H37Ra* (as a source of *Mtb* PAMPs), in the context of NLRP3 inflammasome and caspase-1 activation. We found that, in fluvastatin-untreated PBMCs, macrophages respond to the bacteria by forming granulomatous structures that scarcely activate the NLRP3 inflammasome and caspase-1, resulting in low production of IFNγ. Unexpectedly, the drug induces a generalized conversion of monocytes/macrophages of PBMCs into foamy cells overproducing NLRP3 and ASC in the absence of PAMPs, revealing new cholesterol metabolism/inflammation connections [34]. Based on this, we propose the existence of a cholesterol metabolism/inflammation integrated circuit regulating the activation of the NLRP3 inflammasome, which could explain some autoinflammatory processes. In response to the bacteria, these foamy macrophages did not form compact structures, but degraded the bacilli, markedly exacerbated caspase-1 activation and elicited a potent IFNγ/cytotoxic response. The effect of the drug on IFNγ production was confirmed in vivo by performing a tuberculin test on rabbits either receiving or not receiving fluvastatin therapy, and immunized with inactivated *Mtb H37Ra*. This work offers a useful experimental model to study the events that initiate the formation of granulomas and the role that FMs play in it, also opening new possibilities in the design of alternative therapies to fight not only tuberculosis, but other infectious diseases.

## 2. Materials and Methods

### 2.1. Mtb Strain Selection 

We selected *Mtb H37Ra* as a source of *Mtb* PAMPs since this strain lacks virulence factors and shares most of its membrane proteins with its virulent counterpart *Mtb H37Rv*, conferring on it the capacity to elicit a similar cytokine profile [35,36]. Furthermore, differences in virulence between these strains have been attributed to their different abilities to interfere in the lipid metabolism of macrophages, regardless of their immunogenicity [37]. In addition, as heat-killed or living *Mtb H37Ra* bacilli are equally effective inducers of a granulomatous response [38], we used an inactivated bacteria in order to differentiate immune effects mediated by the drug from those derived from a reduction in the availability of cholesterol by the bacteria. 

### 2.2. Isolation of Peripheral Blood Mononuclear Cells (PBMC) and Culture Conditions 

PBMCs were isolated from blood buffy coats from de-identified samples (without any direct or indirect personal identifiers) of healthy blood donors, provided by the Hematology Service from Hospital La Paz and from Centro de Transfusiones de la Comunidad de Madrid (Madrid, Spain), in accordance with Spanish legislation (*BOE-A-2007-12945, Ley 14/2007, de 3 de julio, de Investigación Biomédica*). Protocols in the study were approved by the Hospital Universitario Ramón y Cajal Ethics Committee (approval code 188/09) in accordance with national and international guidelines. Buffy coats diluted 1:10 in sterile phosphate-buffer saline (PBS) were layered onto Lymphoprep (a Ficoll medium at density 1.077 g/mL, from Nycomed, Oslo, Norway), to isolate PBMCs following the method of Boyum et al. [39]. PBMCs were then re-suspended at a final density of 2 × 10^6^ cells/mL in RPMI 1640 (Gibco, Life Technologies, Carlsbad, CA, USA) supplemented with 10% heat inactivated fetal calf serum, 2 mM L-glutamine, 100 U/mL penicillin, 100 U/mL streptomycin, and 10 μg gentamycin. Aliquots of 2 mL were seeded on 12-well cell culture plates. Half of the wells were supplemented with 5 μM fluvastatin (Novartis Pharmaceutical, Basel, Switzerland) dissolved in dimethyl sulfoxide (DMSO) to obtain a final concentration of 0.04% in the incubation medium. In the remaining wells we added the same amount of DMSO. Plates were incubated at 37 °C in a humidified atmosphere containing 5% CO_2_ in air for 15h and, subsequently, half the wells of each condition were stimulated with 25 μg/mL of heat-inactivated *Mtb H37Ra* (Difco, Sparks, MD, USA), and the incubation was resumed for an additional 24 h period. We analyzed the evolution of the cells in the different cultures as follows: adherent cells from each well were scraped and harvested together with cells in suspension at 1, 4, 8, 12, and 24 h after immune stimulation, which corresponded to 16, 19, 21, 23, 25, 27, and 39 h of incubation of both control and fluvastatin-treated cells. After that, aliquots of 100 μL from each one of the collected samples were cytocentrifuged onto microscope slides (Flex Immunohistochemistry microscope slides, DakoCytomation, Dako, Glostrup, Denmark) by using a Shandon Cytospin 4 cytocentrifuge (Thermo, Runcorn, UK). Cytospin preparations were stained with May–Grünwald-Giemsa solutions (Merck, Rahway, NJ, USA) and analyzed under optical microscopy. 

### 2.3. Transmission Electron Microscopy 

PBMCs were isolated and incubated under the conditions described above. Thereafter, cells were processed for TEM following the method of Reynolds et al. [40]. Briefly, at the end of the incubation, cells from the different conditions were fixed in 2% glutaraldehyde 4% paraformaldehyde in PBS overnight, post-fixed in 1% osmium tetroxide in water for 1h and dehydrated through a series of ethanol solutions (30%, 50%, 70%, 95%, and 100%). After the last dehydration step, samples were incubated in a series of 2:1, 1:1, and 1:2 ethanol and EPON resin mixture and finally embedded in EPON resin at 60 °C for 48 h. Ultrathin sections (50–60 nm) were obtained using a diamond knife (Diatome, Hatfield, PA, USA) in an ultramicrotome (Leica Reichert Ultracut S, Heidelberg, Germany) and collected on 200-mesh copper grids. The sections were counterstained with 2% uranyl acetate in water for 20 min followed by a lead citrate solution for 10 min. Samples were analyzed using a transmission electron microscope (Jeol Jem1010 (100Kv) Tokyo, Japan), equipped with a digital camera (Gatan SC200, Pleasanton, CA, USA).

### 2.4. Immunolocalization of NLRP3 and ASC

Cultures were prepared as above and at the end of the incubation period, the cells were re-suspended. Aliquots of 100 μL of cells from each condition were cytocentrifuged onto microscopy slides and then immersed in 4% paraformaldehyde for 20 min at room temperature. After three washes with PBS, cells were permeabilized with 0.05% Triton X-100 in PBS for 5 min at 4 °C, and washed three times in PBS. Thereafter, slides were incubated in a blocking solution (5% normal goat serum in PBS) for 30 min at room temperature, to prevent unspecific binding of specific antibodies. In the next step cells were incubated overnight, at 4 °C, with mouse monoclonal anti-NLRP3 antibody (Ag 20B-0014-C100, Adipogen Corp., San Diego, CA, USA) and rabbit polyclonal anti-ASC antibody (sc 22514R, Santa Cruz Biotechnology Inc., Dallas, TX, USA) diluted at 1/200 and 1/50, respectively, in 0.5% normal goat serum and 0.001% Triton X-100 in PBS. After primary antibody incubation, cells were washed three times with PBS and then incubated for 45 min at room temperature with fluorescent-conjugated secondary antibodies diluted 1/400 in 0.5% normal goat serum in PBS. The following secondary antibodies were used: Alexa Fluor-568 goat anti-mouse IgG, and Alexa Fluor-488 goat anti-rabbit IgG (both from Molecular Probes, Eugene, OR, USA). Cell nuclei were stained with 3 × 10^−5^ M Hoechst 33342 (Sigma-Aldrich, St. Louis, MO, USA) incorporated to an aqueous mounting medium containing 1 mg/mL p-phenylene diamine and 90% glycerol in PBS. Fluorescent immunostained images were acquired by using a Nikon Eclipse E 400 microscope (Nikon Corporation, Tokyo, Japan).

### 2.5. FAM-FLICA Assay 

PBMCs were isolated and incubated for a period of 15 h plus 24 h, under identical conditions described in Section 2.2. Next, we localized active caspase-1 in living cells by performing a FAM-FLICA (Fluorescent Labeled Inhibitor of Caspases) assay (ImmunoChemistry Technologies, Bloomington, MN, USA). The assay requires the use of DMSO to dissolve the fluorescent inhibitor probe FAM-YVAD, but high concentrations of this solvent cause cells damage and activate caspase-1 [41]. To avoid this, FLICA was dissolved by adding only 5 μL DMSO to each vial and then diluted with 245 μL of PBS (30X working solution). At the end of the incubation, 10 μL of this solution was added to 300 μL of cell suspension (containing 6 × 10^5^ cells) from each condition, and the tubes were gently flicked. The cells were incubated at 37 °C for 1 h protected from light. To ensure an even distribution of the substrate, cells were gently re-suspended every 15 min. After that, cells were centrifuged at 200× *g* for 5 min, the pellets were washed twice in 1.5 mL of PBS with 0.04% DMSO and finally were re-suspended in 300 μL of PBS. Thereafter, aliquots of 50 μL and 100 μL of cells in suspension from each condition were cytocentrifuged onto microscopy slides as mentioned above. Cell nuclei were stained with 3 × 10^−5^ M Hoechst 33342 (Sigma-Aldrich, St. Louis, MO, USA) and incorporated into an aqueous mounting medium containing 1 mg/mL p-phenylene diamine and 90% glycerol in PBS. Cytospin preparations were then analyzed under fluorescence microscopy using a Nikon Eclipse E 400 microscope (Nikon Corporation, Tokyo, Japan).

*Mtb* emits autofluorescence that can be used as a tool for detection of the bacilli in biological samples [42] but, in our study, green autofluorescence interferes with specific fluorescence emitted by FAM-YVAD-FMK. To analyze the results of the assay, we merged red and green autofluorescence. In this way, we could localize active caspase-1 (green fluorescence) and visualize simultaneously the bacteria (as merged yellow fluorescence). To prevent autofluorescence removal during washed procedures, we used a detergent-free washing solution, after evaluating that it did not change assay specificity. It is important to say that such colocalization could not be performed using Ziehl-Neelsen acid-fast stain because solvents used in this technique suppress specific ligand fluorescence. On the other hand, immunolocalization of the bacteria requires cell permeabilization and washes that could affect FAM-FLICA specificity.

### 2.6. Dynamics of Caspase-1 Activation 

A fluorimetric cleavage micro-assay, previously described by us [43], was used to evaluate caspase-1 activation over time. PBMCs from the different conditions were collected at 1, 4, 6, 8, 10, 12, and 24 h after *Mtb H37Ra* stimulation. At each time, adherent cells were scraped, mixed with the cells in suspension, and centrifuged. The collected pellets were lysed via three consecutive freeze-thaw cycles in lysis buffer (25 mM HEPES pH 7.5, 5 mM EDTA, 5 mM MgCl_2_, 0.1% Nonidet P-40, supplemented with 1 mM PMSF, 1 µg/mL aprotinin and 50 µg/mL antipain), using 50 µL for every 12 × 10^6^ cells. Then, lysates were centrifuged at 10,000× *g* for 15 min at 4 °C and the supernatants collected for protein concentration determination. For each sample, a total of 100 mg of protein in 50 mL of lysis buffer was added to 175 mL of reaction solution (25 mM HEPES pH 7.5, 5 mM EDTA, 5 mM MgCl_2_, 22.9% glycerol, 0.15% CHAPS, 11.5 mM DTT, and 175.5 mM NaCl). The prepared samples were placed in microtiter plates Fluoronunc F16 black polysorp (Nalge Nunc International, Rochester, NY, USA), in order to incubate them at 37 °C for 2h in the presence of 100 µM Ac-WEHD-AMC (Calbiochem, San Diego, CA, USA), a specific substrate for caspase-1 that emits fluorescence after its enzymatic cleavage by this protease. Thereafter caspase-1 activity was quantified in each condition by measuring the emitted fluorescence in a fluorometric plate reader (Spectrafluor, Tecan, Männedorf, Switzerland), using an excitation wavelength of 380 nm and an emission wavelength of 465 nm. The specificity of the reaction was assessed by adding equimolar amounts of AC-YVAD-CHO (Bachem, Bubendorf, Switzerland), a specific caspase-1 inhibitor, obtaining a 95% inhibition of the reaction. 

### 2.7. Kinetic Analysis of Cytokines Emission 

We evaluated the production over time of IL-1β, IL-18, IL-10, and IL-12, by cells maintained at the different incubation conditions of this study. We followed the same experimental protocol as for caspase-1 activation dynamics but in this case, we collected cell supernatants. Cytokine quantification was performed by Enzyme-Linked Immunosorbent Assay (ELISA), using commercially available kits and following the manufacturer’s indications. For IL-1β and IFNγ determination, different kits were used and were supplied by R&D Systems (Minneapolis, MN, USA), and Genzyme (Cambridge, MA, USA), as well as ELI-PAIR kits from Diaclone Research (Besançon, France). ELISA kits for IL-18 were supplied by MBL International Corporation (Woburn, MA, USA). ELISA kits for IL-12 and IL-10 were supplied by R&D Systems and by Diaclone Research.

### 2.8. Lactate Dehydrogenase (LDH) Determination

To measure cell death occurring under the different incubation conditions along the incubation period, we prepared cultures following the same experimental protocol as for the kinetic studies described above, but using phenol red-free RPMI 1640 (Gibco, Life Technologies, Carlsbad, CA, USA). After immune stimulation, supernatants were collected at the same intervals as above, and LDH activity was measured by using the CytoTox 96 Kit (Promega, Madison, WI, USA), according to the manufacturer’s instructions. Cytotoxicity was expressed as a percentage of the maximum LDH release in control conditions. 

### 2.9. Contribution of IL-12, IL-1β, and IL-18 to IFNγ Production

PBMCs were placed on 12-well plates and either treated or not treated with 5 μM fluvastatin as above. To different wells from both conditions, we added specific antibodies against IL-1β (1.25 μg/mL, C20 sc 1250), IL-18 (1 μg/mL, C18 sc-6177) or IL-12 p35 (0.5 μg/mL, C19 sc-1280), all of them from Santa Cruz Biotechnology (Dallas, TX, USA. To other wells we added the specific inhibitor of caspase-1, Ac-YVAD-CHO (Bachem AG, Bubendorf, Switzerland) at a concentration of 100 μM. Thereafter, plates were incubated for 15 h; then, half of the aliquots from each condition were stimulated with 25 μg/mL of heat-inactivated *Mtb H37Ra,* and the incubation was resumed for a further 24 h. At the end of the process, supernatants were collected and IFNγ quantified by ELISA (R&D Systems, Minneapolis, MN, USA).

In an independent study, PBMCs were either treated or not treated with 5 μM fluvastatin as above and supplemented with recombinant human IL-12 (Pepro Tech, London, UK) at the following concentrations: 0.01, 0.02, 0.10, 10 or 100 ng/mL. Thereafter, cells were incubated according to our basic experimental protocol. At the end of the process, supernatants were collected for IFNγ quantification by ELISA (R&D Systems, MN, USA).

### 2.10. Intracellular IFNγ Determination by Flow Cytometry 

For surface staining, PBMCs from the different conditions were incubated at a density of 8 × 10^5^ cells/100 μL for 30 min with mixtures of fluorochrome-conjugated mAbs (or isotype-matched controls). The following conjugated mAbs were used: anti-human CD3 APC, anti-human CD8 PerCP, and anti-human CD56 PE, all obtained from Becton Dickinson (BD Biosciences, CA, USA). Then, cells were fixed and permeabilized using the Intra Stain kit from Dako (Glostrup, Denmark). We avoided the use of protein transport inhibitors in our cytometry study because of its interference with fluvastatin [44]. Once permeabilized, the cells were washed and incubated for 30 min with anti-human IFNγ conjugated to FITC from Becton Dickinson (BD Biosciences, San Diego, CA, USA). Four-color analyses were performed using a FACSCalibur flow cytometer from Becton Dickinson (BD Biosciences, CA, USA), with 10,000 events collected for each tube. The cells were gated using the forward scatter (FSC) and side scatter (SSC) to select the lymphocyte population. The intracellular production of IFNγ was studied on three different cells types: CD3^+^CD8^+^ cells, CD3^+^CD8^−^ cells, and CD3^−^CD56^+^ cells. The analysis was performed using the CellQuest Pro TM Software from Becton Dickinson (BD Biosciences, San Diego, CA, USA). 

### 2.11. Rabbit Treatment and Immunization 

Five New Zealand white rabbits (2 months old and weighing 2 Kg) were fed with a standard chow diet and another five were fed with the same food but supplemented with 2 mg/kg/day of fluvastatin (Novartis Pharmaceutical, Basel, Switzerland) as follows: the drug was dissolved in acetone and mixed with food, allowing the solvent to evaporate from the mixture before offering it to animals. After 15 days of treatment, all rabbits were immunized by intramuscular inoculation of 5 mg of heat-inactivated *Mtb H37Ra* emulsified in incomplete Freund adjuvant (Difco, Sparks, MD, USA). Fifteen days later rabbits were boosted with the same doses. Forty-five days later we performed a tuberculin skin test (PPD Evans, AJ Vaccines, Copenhagen Denmark), by injecting an intradermal dose of 2 tuberculin units (TU- 0.1 mL). The reaction was read 72 h after injection by measuring with a caliper the skin fold thickness in the area of inoculation. Animal experimentation was performed in accordance with Spanish legislation on the protection of animals used for experimentation and other scientific purposes (Real Decreto 22311988), that was in force when the study was carried out. All the experimental protocols were reviewed and approved by a scientific committee from Hospital Universitario Ramón y Cajal. 

### 2.12. Statistical Analysis

Results are expressed as mean ± SEM from several independent experiments indicated in each figure. For parametric data, a Student’s *t*-test, or one or two-way ANOVA followed by the Newman–Keuls multiple comparison test were performed. Differences were considered significant when *p* ≤ 0.05. Statistical analysis was performed with Graph Pad Prism 6 software (La Jolla, CA, USA).

## 3. Results

### 3.1. Fluvastatin Converts Macrophages to Foamy Cells Unable to Form Granulomatous Structures in Response to Inactivated Mtb H37Ra

We started this study by analyzing under light microscopy the effects of fluvastatin on PBMCs and the response that these cells elicited against inactivated *Mtb H37Ra*. PBMCs were either treated or not treated with the statin for 15 h. Next, we added the bacteria to half of the wells from each condition, and resumed the incubation for another 24 h. At the end of this incubation, macrophages from drug-untreated cultures (control condition) showed numerous small vacuoles (Figure 1A), but in fluvastatin-treated cultures most macrophages became highly vacuolated rounded cells, with an eccentric and concave nucleus. Many of these macrophages were tightly attached to lymphocytes and/or monocytes (see Figure 1B), which here we will call “small cells”. In PBMCs exposed only to the bacteria, macrophages formed compact aggregates enclosing central cores (Figure 1(C1)), similar to granulomas generated by viable *Mtb H37Rv* in PBMCs from healthy donors [45]. Thus, we will refer to these aggregates as granuloma-like structures (GLSs). In this condition, a few macrophages became highly vacuolated cells (Figure 1(C2)) such as those shown in Figure 1B. In fluvastatin-treated cultures, the stimulation with the bacteria did not generate GLSs; instead, vacuolated macrophages formed cellular aggregates surrounding smaller central cores (Figure 1D). All these vacuolated macrophages presented morphological characteristics of FMs, which are a group of scavenger cells that perform homeostatic functions in human tissues and fluids [46,47]. Appendix A shows the similarity between the vacuolated macrophages generated in this work and conventional FMs that remove erythrocytes (erythrophages) in a hemorrhagic pleural effusion sample prepared for diagnostics at Hospital Universitario Ramón y Cajal. Here, we will refer to vacuolated macrophages observed in cultures only exposed to the bacteria as FMs and to macrophages transformed by fluvastatin as FFMs.

A temporal evolution study revealed that GLSs started to be formed just after adding the bacteria. During the first hour of immune stimulation, we observed a small number of formations in which two or a few more macrophages surrounded small amorphous masses (Figure 2(A1,A2)). A little later, these formations evolved towards larger amorphous masses surrounded by dismantled nuclei from the initial macrophages and a few other macrophages, including FMs, which arrived later (Figure 2(A3)). This result indicates that some of the macrophages initiating the formation died by necrosis and the debris generated became mixed with the accumulated material. Thus, we will refer to these as necrotic cores or necrotic areas. We could not determine the number of macrophages that died in this action, but it seemed to be very small. In the following hours a high number of macrophages were quickly incorporated into these formations enclosing them in the core of small GLSs (Figure 2(A4)). During the rest of the incubation period, many more macrophages were incorporated to GLSs, which increased in size. These newly recruited macrophages became compacted over time and the structures stained more intensely in blue. As observed in Figure 2(A5) the central core also grew over time. The number of FMs in these cultures increased progressively and after 4h of adding the bacteria they represented 2-4% of total macrophages. These cells were incorporated into GLSs, impeding the final count of the transformed macrophages. At the end of the incubation period, most of the GLSs were highly compacted, intensely stained, and showed few FMs (Figure 2(A5)). However, a reduced number of GLSs enclosed smaller necrotic cores and had numerous FMs (Figure 2(A6)). The formation of GLSs was not a synchronized process and we were able to observe incipient granulomas at all times of the incubation period. In cultures treated with fluvastatin and exposed to the bacilli, a few FFMs surrounded similar amorphous masses (Figure 2(B1,B2)) that barely grew over time. At the end of the incubation period, they formed loose cellular aggregates (Figure 2(B3–B5)), but occasionally we saw some more compacted structures with fewer FFMs (Figure 2(B6)). In the absence of bacteria, neither the untreated nor the fluvastatin-treated cultures formed compacted cellular aggregates (Appendix A). These findings indicate that the drug not only deeply changes the morphology of macrophages, but also prevents the activation of a granulomatous response against the bacteria.

### 3.2. Ultrastructural Study of Macrophages

To define more precisely the changes that occurred in macrophages from the different conditions, we performed a transmission electron microscopy study. Compared to macrophages from control cultures (Figure 3A), most macrophages exposed to inactivated *Mtb H37Ra* increased in size, contained a large and elongated nucleus and showed ruffled membranes with projections ranging from small and extended pseudopodia to voluminous and irregular masses (Figure 3(B1)). These macrophages captured the bacteria in phagosomes that barely degraded them, and interlocked with each other through a labyrinthine network of membrane projections (Figure 3(B2,B3)). All these cellular characteristics are coincident with those of epithelioid cells from tuberculous granulomas [48] and support that heat-killed or living *Mtb H37Ra* are equally effective inducers of these granulomas [38]. In the same culture, a few macrophages preserved their size and showed large cytoplasmic vacuoles containing non-identifiable particles and debris (Figure 3(B4–B6)). These cells emitted numerous short membrane protrusions that captured material from their surrounding microenvironment, forming a lattice of small vacuoles at the edge of the cell. Some of these macrophages enclosed non-degraded bacteria in phagosomes (Figure 3(B7)), but they also showed other bacilli captured together with debris and delivered into vacuoles, where they interacted with vesicular organelles with a size and morphology similar to secondary lysosomes described by Armstrong and Hart [49]. Furthermore, signs of bacterial degradation at the contact points of this interaction were visible, indicating that these formations were in fact, degrading organelles. This type of phagocytosis, resembling macropinocytosis, is associated with the cleaning of necrotic debris at sites of injury and inflammation [50]. Thus, we consider that these vacuolated macrophages are FMs performing scavenger tasks.

FFMs resulting from fluvastatin treatment had numerous large vacuoles (Figure 3(C1)), and emitted short membrane protrusions that engulfed non-identifiable material from their surrounding microenvironment (Figure 3(C2)). However, as in this condition immune debris was barely generated, and most macrophages were able to remove it, their vacuoles scarcely contained detritus and were apparently empty. A few of these FFMs contained vesicular organelles within their vacuoles (Figure 3(C3)) such as those shown in Figure 3(B8). We appreciated a tight association between the cytoplasmic membranes of FMs and their attached cells (Figure 3(C4,C5)). Another feature of these macrophages was that they frequently subdivided vacuoles by emitting filopodia-like projections (Figure 3(C6)). In fluvastatin-treated cultures that were exposed to the bacteria, FFMs were not converted to epithelioid cells and only emitted small membrane protrusions. These activated FFMs showed few vacuoles and their cytoplasm was filled with debris and a high number of vesicular organelles, both intermingled with filopodia-like projections (Figure 3(D1)). Similar organelles were present at the periphery of these macrophages (Figure 3(D2)). Like FMs, these FFMs captured some bacteria in phagosomes (Figure 3(D3)) and engulfed others along with debris through a similar phagocytic process, but delivered them into a cytoplasm filled with organelles that could degrade them (Figure 3(D3,D4)). Supporting this possibility, preserved bacterial cells were infrequently visualized in these cells, although their cytoplasm contained dark spots that, due to their morphology and size, could correspond to partially digested bacteria (Figure 3(D5,D6)). Many of these cells presented numerous lipid bodies (Figure 3(D5)) similar to those observed in macrophages infected with the bacillus of Calmette–Guerin [51]. 

### 3.3. FFMs Overproduce NLRP3 and ASC and the Bacteria Promotes the Co-Localization of Both Proteins 

To analyze if statins could interfere in the formation of NLRP3 inflammasomes, we studied by immunofluorescence the distribution and colocalization of ASC and NLRP3 in cells from the different conditions. In control cultures, a few macrophages showed some ASC inside the nucleus, and some NLRP3 distributed as small dots at perinuclear locations (Figure 4). In cultures exposed to the bacteria, most macrophages in GLSs had augmented ASC in their nuclei without evidence of an increase in NLRP3, although it appeared concentrated as a single dot near the nucleus (Figure 4, Mtb H37Ra upper row), an event considered as one of the initial steps for the assembly of the inflammasome [52]. In this condition, some FMs (cells with an eccentric and concave nucleus) formed ring-shaped aggregates of NLRP3, and partially moved ASC from the nucleus to co-localize with them (Figure 4, Mtb H37Ra bottom row). Unexpectedly, FFMs showed a considerable increase in NLRP3 in their cytoplasm, forming several ring-shaped aggregates such as those seen in FMs, despite the absence of exogenous PAMPs (Figure 4, fluvastatin row). The number of these formations was highly variable among FFMs from the same culture and in cultures from different donors. ASC was also increased in the nuclei of all these cells, but co-localization with NLRP3 was only observed in some of them. In fluvastatin-treated cultures exposed to the bacteria, FFMs massively released ASC from the nucleus to partially co-localize with NLRP3 in numerous ring-shaped aggregates (Figure 4, fluvastatin + Mtb H37Ra row). Furthermore, these macrophages released particles where ASC and NLRP3 partially co-localized (Figure 4, fluvastatin + Mtb H37Ra), indicating that they could assemble at the periphery of organelles that can be released out of the cell.

### 3.4. Fluvastatin Exacerbates Caspase-1 Activation in Response to Mtb PAMPs

Active caspase-1 was localized in cells by performing a FAM-FLICA assay. In control cultures, there was specific green fluorescence as dots in broken cells, inside dismantled nuclei or, as observed by others [53], forming large extracellular aggregates (Figure 5(A1,A2)). In fluvastatin-treated cultures, some FFMs showed specific fluorescence delineating either a ring at a perinuclear position, or several rings or arcs dispersed in the cytoplasm (Figure 5(B1,B2)), such as the NLRP3 aggregates seen in Figure 4. Unexpectedly, in PBMCs exposed to the bacteria, a small subset of macrophages captured large amounts of bacilli emitting red and green autofluorescence (see the Section 2). Merging both fluorescences, the bacteria were delineated in yellow but we could not detect specific green fluorescence, indicating the absence of active caspase-1 in these macrophages (Figure 5(C1–C3)). A few of these macrophages associated to accumulate these bacilli in a central core (Figure 5(C4)), comparable to the amorphous masses shown in Figure 2A. In small incipient GLSs (Figure 5(C5)), other macrophages, with much lower bacterial load and without active caspase-1, surrounded the accumulated bacteria. In larger GLSs (Figure 5(C6)), numerous macrophages were tightly compacted, compressing the accumulated bacilli in their central core. These results and those presented in Section 3.1 indicate that only a small number of macrophages die to accumulate huge numbers of bacteria in the core of the structure. In this condition, only a few macrophages, which could be FMs due to their morphology, showed specific green fluorescence enclosed in a few round formations (Figure 5(C7,C8)) with a similar size and cytoplasmic distribution to that of NLRP3/ASC aggregates (Figure 4, *Mtb H37Ra* lower row). These coincidences and the fact that specific fluorescence accumulated in rings at their edges (Figure 5(C7)), indicate that the NLRP3 inflammasome activates caspase-1 at the periphery of these formations and that, once active, the enzyme is released inside them. In fluvastatin-treated cultures exposed to the bacteria, most FFMs markedly increased the number of round formations enclosing active caspase-1 (Figure 5(D1,D2)) at the same proportions as NLRP3/ASC ring-shaped aggregates (Figure 4). These data suggest that FFMs assembled the inflammasome and activated caspase-1 through the same pathway as FMs, but that this process was exacerbated in the former. Most of these activated FFMs did not show intact bacteria, but had bacilli remnants in the same formations as active caspase-1, sometimes forming yellow arcs at their edges (Figure 5(D1) and Appendix A). Such co-localization and the fact that these formations had a similar number, size, and cellular distribution to those of degrading organelles (Figure 3D), could indicate that both correspond to the same structure. In this condition, a few macrophages preserved their ability to capture large numbers of bacteria (Figure 5(D2)), also accumulating them in a central core. Some of these macrophages seem to harbor the bacteria in their broken nuclei (Figure 5(D3)). None of these macrophages activated caspase-1. Close to them, other FFMs showed numerous round formations with large amounts of specific green fluorescence unmasking a faint red autofluorescence emitted by small fragments of the bacilli (Figure 5(D4), white arrow). These results indicate that FFMs fully degraded the bacteria accumulated by the former, supporting TEM observations. In Appendix A, additional images evidencing that FM degrade the bacteria are shown.

We also quantified caspase-1 activity in lysates from cells collected from each condition at different times of incubation. Results are presented in Figure 6A. In agreement with FAM-FLICA results, the control condition showed a basal caspase-1 activity sustained over time. In cultures exposed to the bacteria or treated with fluvastatin, this basal activity tends to increase at the end of the incubation period but it did not reach statistical significance. When bacteria and drug were combined, caspase-1 activity significantly increased from 10 h of immune stimulation onwards.

### 3.5. Fluvastatin Exacerbates the Cellular Response against Inactivated Mtb H37Ra

After stimulation with the bacteria, we quantified the levels of IL-1β, IL-18, IL-10, IL-12 and IFNγ in culture supernatants collected over time. As shown in Figure 6A,B, the levels of these cytokines were undetectable in supernatants from control cells. In cultures treated with fluvastatin or with the bacteria alone, IL-1β and IL-18 production was not statistically different from that of controls (Figure 6A). In cultures stimulated with the bacteria, IL-10 and IFNγ increased progressively with time, reaching statistical significance vs. the control and fluvastatin conditions at the end of the incubation period, while IL-12 increased significantly after 10 h of immune stimulation, and peaked two hours later (Figure 6B). When we combined fluvastatin and bacteria, the amount of IL-1β significantly increased at 6 h of co-incubation, later reaching very high values. IL-18 levels augmented starting 10 h after adding the bacteria (Figure 6A). On the other hand, IL-10 was significantly reduced without affecting IL-12 production. Nonetheless, IFNγ levels were markedly increased after IL-12 peaked (Figure 6B). Caspase-1 inhibition by its specific substrate YVAD, as well as the addition to the cultures of neutralizing antibodies targeting IL-1β or IL-18, prevented this peak of IFNγ (Figure 6C). Furthermore, it was necessary to supplement the cultures with high concentrations of recombinant human IL-12 to slightly increase the levels of IFNγ induced by the bacteria (Figure 6D). These results indicate that, in response to inactivated *Mtb H37Ra*, IL-12 is required for IFNγ production, but that the levels of ILs processed by caspase-1 determine the amount of cytokine produced.

**Figure 6 cells-13-00536-f006:**
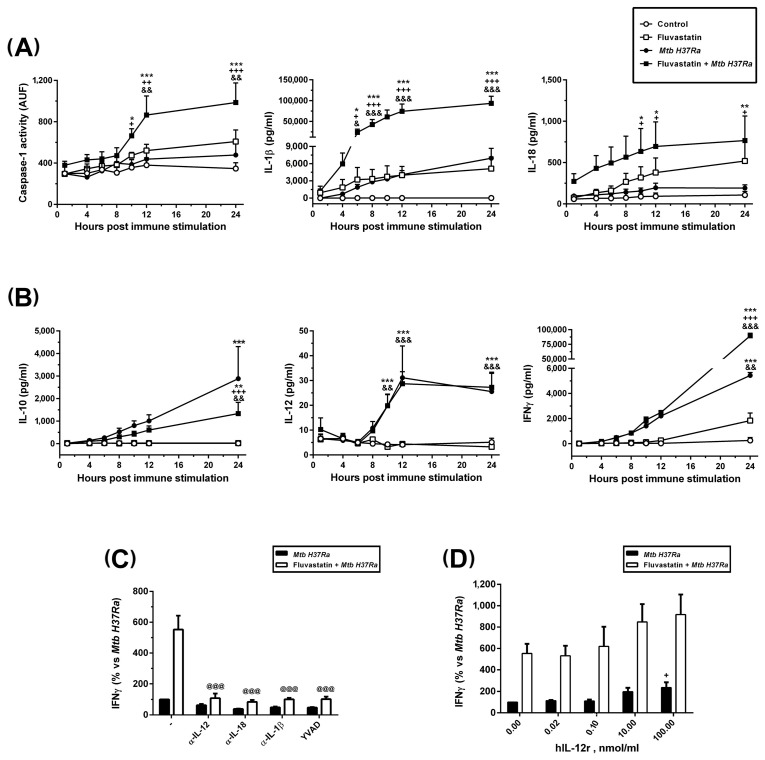
Dynamics of caspase-1 activation and production of cytokines. (**A**) Time course of caspase-1 activity in cell lysates expressed in arbitrary units of fluorescence (AUF), and IL-1β and IL-18 concentrations in the media after immune stimulation. (**B**) Time course of the release of IL-10, IL-12 and IFNγ to the media. (**C**) Effect of neutralizing antibodies independently targeting IL-12, IL-18 or IL-1β, and the caspase-1 inhibitor YVAD on IFNγ release induced by the bacteria in untreated (black bars) or fluvastatin-treated cultures (white bars). (**D**) Dose–response effect of human recombinant IL-12 on the production of IFNγ induced by the bacteria alone (black bars), or in combination with the bacteria (white bars). In (**A**,**B**), results represent the mean ± SEM of 6 independent experiments. In (**C**,**D**), values are normalized by IFNγ produced by fluvastatin-untreated PBMCs in the presence of bacteria (100%) and represent the mean ± SEM of 4 independent experiments. * *p* ≤ 0.05; ** *p* ≤ 0.01, *** *p* ≤ 0.001 vs. control, ^+^ *p* ≤ 0.05; ^++^
*p* ≤ 0.01, ^+++^
*p* ≤ 0.001 vs. *Mtb H37Ra*, ^&^ *p* ≤ 0.05; ^&&^ *p* ≤ 0.01, ^&&&^ *p* ≤ 0.001 vs. fluvastatin, **^@@@^** *p* ≤ 0.001 vs. fluvastatin + *Mtb H37Ra* (one-way (**C**,**D**) or two-way ANOVA (**A**,**B**) +Newman–Keuls).

Next, we analyzed the subsets of lymphocytes producing IFNγ by flow cytometry. In cultures exposed to the bacteria, either treated or not treated with fluvastatin, the forward scatter versus side scatter dot plot showed only a gate of lymphocytes, indicating that most macrophages had been incorporated to GLS or to cellular aggregates. Despite the impossibility of using protein transport inhibitors in this study (see Materials and Methods), we detected in this gate a low number of CD3^+^/CD8^+^, CD3^+^/CD8^−^ lymphocytes and CD3^−^/CD56^+^ (NK) T-cells that produced IFNɣ in response to the bacteria. The number of positive cells increased in all these populations in PBMCs from the same donors treated with fluvastatin and bacteria (Figure 7A). Since NK and CD3^+^/CD8^+^ cells are cytotoxic, we measured LDH activity in supernatants to evaluate cellular death. In control cultures, basal LDH activity was maintained through the incubation period, and it was not significantly increased by the exposure to the bacteria despite the fact that GLSs seemed to be initiated by necrotizing macrophages (Figure 7B). Possibly, the number of necrotizing macrophages was so low that the cytotoxic test is not sensitive enough to differentiate small increases in LDH from basal. Fluvastatin treatment increased this activity, but when combined with the bacteria it notably augmented (Figure 7B).

We also evaluated the effect of fluvastatin on IFNɣ production in response to bacteria in vivo by performing a tuberculin test in rabbits immunized with the same inactivated *Mtb H37Ra*. As shown in Figure 7C, rabbits immunized with the bacteria were tuberculin-negative. However, the same protocol of immunization in combination with fluvastatin therapy, markedly increased the size of the induration area (Figure 7C). 

## 4. Discussion

The regulation of the early immune events that determine the resolution or perpetuation of tuberculosis is presently seen as a therapeutic target to control the disease, but studying in vivo this early interaction has important technological limitations [54]. Herein we observed that, in response to inactivated *Mtb H37Ra*, monocytes/macrophages from healthy donors polarize their functions towards different phenotypes that cooperate with each other to form granulomatous structures. This is a stepwise process, which, as we will argue below, mimics early stages in the formation of tuberculous granulomas, providing valuable information on this regard.

It is generally agreed that, before tuberculous granulomas begin to organize, virulent *Mtb* strains accumulate and proliferate in necrotic areas, evading immunity [55]. Some authors have proposed that these foci of death are generated by minimally microbicidal macrophages that allow the pathogen to grow intracellularly [3]. These macrophages then die by necrosis releasing viable pathogens into a harmful extracellular space. Paradoxically, the bacilli grow and spread even further in this environment [56,57,58]. In the present work, a small subset of macrophages quickly captures large numbers of inactivated *Mtb H37Ra* to the point of death, also forming necrotic areas where the bacilli accumulate. Given that in our study the bacteria lacks virulence factors and cannot infect macrophages or proliferate inside them, the formation of these areas cannot be ascribed to its virulent activity. We thought that, in healthy people, a subset of macrophages from the blood could be functionally predetermined (or perhaps trained) to detect free mycobacteria and, regardless of their virulence, concentrate them in an inhospitable space to impede their survival. It could represent a previously undescribed strategy of innate immunity to quickly prevent the uncontrolled spread of free mycobacteria to other parts of the body, however virulent *Mtb* strains would evade it because they have learned to survive and proliferate in this space [8]. Herein we also observed that, in only 24 h of immune stimulation, most of the remaining macrophages become epithelioid cells that enclose these necrotic foci in compact GLSs. These results may support that blood-derived macrophages are important players in the initiation of a granulomatous response against *Mtb*, without requiring specific tissue factors [59]. Central cores in GLSs expand over time, suggesting that some necrotizing macrophages continue replenishing necrotic areas with bacteria and debris once the structure is formed. In line with this, in animals superinfected with an isogenic *Mtb* virulent strain, the new infectious bacilli are rapidly transported within some macrophages to the necrotic centers of the preexisting granulomas, where they die and the bacilli proliferate [60]. It could be of interest to study whether a defective response of these necrotizing macrophages could underlie in some cases of miliary tuberculosis [61]. 

Another initial key event influencing the progression of granulomas is the conversion of FMs to LL-FMs by the pathogen. We want to remember that, although the term “foamy macrophage” frequently refers to lipid-laden macrophages [62], pathologists consider them scavenger cells that recover homeostasis in damaged tissues and fluids [46,47]. In agreement, the biogenesis and function of these cells vary across diseases [63,64], where they accumulate not only lipids but other products derived from the engulfed waste [65]. Thus, in tuberculosis the storage lipid by FMs is disease-specific [66] and differs from that of FMs in atherosclerotic lesions although, whether they also differ in their immune function remains unknown [67]. In PBMCs exposed to the bacteria, a subset of macrophages turns into FMs and are quickly recruited to incipient necrotic areas. These cells enclose intact bacilli in phagosomes and emit short protrusions to engulf other bacilli together with debris, to be degraded in large vacuoles with the help of some unidentified vesicular organelles. Thus, we consider that these macrophages become FMs to clean early immune debris generated, as observed in vivo [68]. Overtime, these FMs are incorporated into GLSs without responding to signals for epithelioid transformation, what could explain why these cells accumulate around the caseum in human granulomas [69]. Despite these analogies, since in our study the bacteria are dead, they cannot convert FMs to LL-FM; however our experimental model allows us to study how these cells would respond to *Mtb* PAMPs before being transformed by the pathogen. Here, we found that these activated FMs augment ASC in the nuclei and NLRP3 in the cytoplasm, the latter forming previously undescribed ring-shaped aggregates. In some of these cells, ASC partially leaves the nuclei to colocalize with NLRP3 aggregates and they show active caspase-1 inside previously undescribed round formations similar in number, size, and cytoplasmic distribution to ASC/NLRP3 rings, strongly suggesting that FMs assemble this inflammasome around organelles that retain the enzyme once activated. This pathway for NLRP3 inflammasome and caspase-1 activation has not been referred in other studies [28], but it shows analogy with a mechanism of inflammasome regulation which has not been deeply studied [52]. Unlike FMs, most of the cells forming GLSs hardly augment levels of NLRP3 in response to *Mtb* PAMPs and, although ASC increases, it remains in the nuclei, an event that prevents caspase-1 activation [52]. Consequently, the secretion of IL-1β and IL-18 was not significantly different from that observed in control cultures. Furthermore, these structures also act as active platforms for the release of very low amounts of IL-12 and high quantities of IL-10. This IL pattern is similar to that produced by tuberculous granulomas in vivo, also resulting in the activation of poor IFNγ production [70]. The simplicity of our experimental model offers many advantages to design complementary studies directed to decipher what signals transform macrophages into FM or into epithelioid cells and the role that they play in the formation of these structures.

An unexpected result of the present study is that, in the absence of immune stimulation, fluvastatin massively converts macrophages of PBMCs to foamy cells (FFMs), similar to FMs induced by the bacteria in untreated PBMCs and both clear bacteria along with cell debris through the same process of phagocytosis. These cells also associate with small cells through close intercellular attachments, strongly suggesting the existence of a communication among them that, at least in the case of FFMs, cannot be linked to foreign antigen presentation. These similarities suggest an important participation of cholesterol metabolism in the conversion of resting macrophages to conventional FMs. As schematized in Figure 8, fluvastatin not only inhibits biosynthesis of sterols but also of isoprenoids such as geranylgeranyl-pyrophosphate (GGPP) and farnesyl-pyrophosphate, that modulate numerous cell signaling processes [21] and cytoskeleton organization [22]. Furthermore, cholesterol depletion induced by statins promotes the synthesis of the sterol regulatory element-binding protein 2 (SREBP2) [71], which binds to the SREBP cleavage-activating protein (SCAP), initiating a process aimed at recovering cholesterol homeostasis in the cell [72]. This process has been associated with foam cell formation [73]. Thus, to understand the complex metabolism/immunity interactions that transform macrophages to FMs it would be necessary to carry out complementary studies. 

Of note, despite the absence of immune stimulation, FFMs overproduce ASC in the nucleus and NLRP3 in the cytoplasm, the latter forming numerous ring-shaped aggregates such as those observed in FMs. This result indicates that the inhibition of cholesterol metabolism generates in macrophages an immune alert that mimics signal 1 mediated by *Mtb* PAMPs in FM. We have previously demonstrated that GGPP prevents caspase-1 activation promoted by fluvastatin. Later, other studies evidenced that the inhibition of the mevalonate pathway and mevalonate kinases deficiencies (MKD) lead to a shortage of GGPP that upregulates *NLRP3* expression [74,75,76,77] by compromising geranylgeranylation of as yet unidentified proteins [78,79]. Thus, in the present work a fluvastatin-mediated reduction in GGPP in FFMs would promote *NLRP3* expression, but its elevated production suggests this process could be deregulated. Guo et al. [80] have previously shown in macrophages that the dimer SCAP-SREBP2 forms a ternary complex with NLRP3, thereby integrating the activation of the NLRP3 inflammasome to signals that regulate cholesterol homeostasis [72] (see Figure 8B). In this study, the authors observed increased expression of *HMGCR*, but its role in NLRP3 inflammasome activation was not addressed. We propose that the increased synthesis of this enzyme would restore GGPP levels in the cell to downregulate NLRP3 expression and limit inflammasome assemblage, evidencing the existence of an immunity/cholesterol metabolism integrated circuit regulating inflammation (see Figure 8B). Accordingly, as in FFMs fluvastatin may inhibit de novo synthesized HMG-CoA reductase, these macrophages would not be able to recover GGPP levels and the receptor would accumulate. Supporting the existence of this circuit, macrophages from patients with MKD (unable to phosphorylate mevalonic acid) not only accumulate NLRP3 but also have increased mevalonic acid levels and HMG-CoA reductase activity [78]. Thus, supplementation with exogenous GGPP could help in the management of these patients [81]. 

**Figure 8 cells-13-00536-f008:**
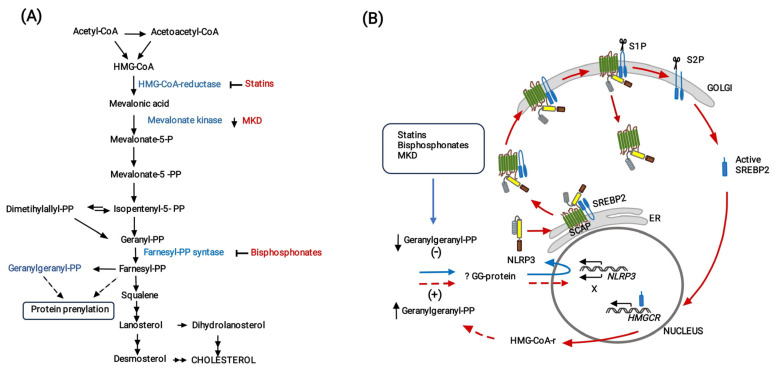
Proposal for a cholesterol metabolism/immunity integrated circuit regulating NLRP3 inflammasome activation in macrophages. (**A**) Schematic representation of the cholesterol biosynthesis pathway. The inhibition of this pathway by statins or bisphosphonates and mevalonate kinase deficiencies (MKD) reduces the synthesis of geranylgeranyl-PP (GGPP). (**B**) In all these cases, reduced geranylgeranylation of an as yet unidentified protein (? GG protein) induces overexpression of *NLRP3* (blue arrow). According to Guo et al. [80] (red arrows), NLRP3 forms a ternary complex with SREBP2 and SCAP that is translocated from the endoplasmic reticulum (ER) to the Golgi apparatus. Two consecutive proteolytic cleavages of SREBP2 by S1P (site 1 protease) and S2P (site 2 protease) generate the active form of the transcription factor, which increases the expression of *HMGCR*, whereas the remaining SCAP-NLRP3 complex is released from the Golgi, optimizing the activation of the inflammasome. We propose that de novo synthesized HMG-CoA reductase allows the cell to restore GGPP levels to downregulate NLRP3 expression, thereby closing (discontinuous red lines) a previously undescribed cholesterol metabolism/immunity integrated circuit involved in the regulation of inflammation.

ASC is also overexpressed in FFMs, most of it being retained in the nucleus, thus preventing inflammasome assemblage [52]. However, in some FFMs, ASC moderately leaves the nucleus and partially colocalizes with NLRP3 and a few of them show specific FAM-YVAD-FMK fluorescence delineating rings with a similar size and cytoplasmic distribution to that of NLRP3. Possibly, in these FFMs the specific ligand binds to an intermediate form of partially active caspase-1, which remains bound to the inflammasome, as reported by others [82]. In agreement with this, there is an increasing trend in caspase-1 activity in cellular lysates from this condition as compared to control cells. The mechanism for fluvastatin-induced synthesis of ASC is unknown, but it evidences the existence of additional cholesterol metabolism/inflammation links. New studies directed to decipher how statins regulate ASC and NLRP3 expression could help to understand other inflammatory pathologies linked to a deregulated cholesterol metabolism [25,83]. 

All the effects exerted by fluvastatin on healthy macrophages deeply affect their response to *Mtb H37Ra*. The drug does not impede that some macrophages accumulate the bacilli in necrotic areas. However, FFMs do not respond to signals for epithelioid transformation, and thereby they do not form GLSs, but instead they capture and destroy the accumulated bacteria. In response to *Mtb* PAMPs, FFMs markedly increase the number of NLRP3 ring-shaped aggregates induced by the drug, providing additional evidence that *NLRP3* expression is unregulated in these macrophages. It has been described that different cellular organelles, which vary between different cells, can act as supports for the trafficking and assembly of the inflammasome components, thus mediating different responses [84]. In our study, FFMs respond to inactivated *Mtb H37Ra* by assembling the NLRP3 inflammasome and activating caspase-1 around numerous organelles that degrade the bacteria and retain the enzyme once it has been activated. The presence of bacilli remnants forming arcs at the edges of these organelles suggests that degradation of bacilli provides inflammasomes with PAMPs to generate a signal 2 that fully activates caspase-1, a possibility not described so far [85]. Comparing the response of FFMs and FMs to the bacteria, we conclude that both activate the same inflammatory pathway but that fluvastatin highly exacerbates it. We consider that this inflammatory response could be a characteristic of FMs that remains unexplored because these macrophages have not yet been studied appropriately. 

A recent study in primates evidences that only those granulomas able to activate T-cells and NK cells for IFNγ/cytotoxic response, can be associated with bacterial clearance, whereas in the others, the bacteria persist and grow [86]. The high levels of IL-1β and IL-18 produced by FFMs against the bacteria activates a potent IFNγ/cytotoxic response just after IL-12 peak, despite its levels remaining very low. This agrees with the fact that IL-12 renders human T-cells highly responsive to IL-1β and IL-18 for IFNγ production by inducing the expression of IL-18 receptors [87]. The fact that YVAD and neutralizing antibodies targeting IL-1β or IL-18 independently prevent this peak of IFNγ supports the importance of caspase-1 in the activation of a protective response against the bacteria [24]. Furthermore, in our previous study [43], this potent inflammatory response promoted by fluvastatin against inactivated *Mtb H37Ra* was prevented by geranylgeraniol, supporting the importance of this isoprenoid in controlling the inflammation mediated by the NLRP3 inflammasome. Our cytometry results involve T-cells and NK cells in this augmented IFNγ response, but we do not exclude the possibility that some “small cells” bound to FMs can contribute to it. 

The fact that in fluvastatin-untreated PBMCs exposed to the bacteria, the same T-cells and NK cells scarcely activate IFNγ production, indicates that healthy people have enough specific immune cells to activate a protective IFNγ/cytotoxic response against Mtb *H37Ra*, but that these cells require to be properly instructed by the infected macrophages. These findings support that granuloma formation limits the effectiveness of adaptive immunity in controlling the bacteria [2]. Since FFMs cannot form GLSs, statins could promote the dissemination of the bacilli. Yet, this possibility seems unlikely because the drug does not prevent macrophages from accumulating the bacilli in central areas, which are then surrounded by numerous FFMs that effectively capture and destroy the accumulated bacteria. Based on our results, we hypothesize that control of *Mtb* infection may require not only generating specific T-cells, but also helping macrophages to activate them correctly. This proposal also reinforces the idea that adaptive immunity against *Mtb* is difficult to improve with current vaccines [88] and ratifies the view of some authors that have questioned the efficacy of vaccination and immunity enhancing strategies, which only mimic the natural immune response to *Mtb*, to control pulmonary tuberculosis [89]. In agreement with our in vitro results, rabbits immunized with inactivated *Mtb H37Ra,* the tuberculin test was negative, but when rabbits received fluvastatin treatment, the same immunization protocol significantly increased the induration area in this test, validating in vivo the potential of statins to increase the production of IFNγ in response to *Mtb* PAMPs. This finding raises the possibility that the tuberculin test may generate false positives or be exacerbated in those patients treated with statins. 

Together, our in vitro and in vivo results suggest that in hypercholesterolemic patients receiving statin therapy, these drugs could help FMs to activate an efficient IFNγ/cytotoxic response that controls the pathogen at the earliest stages of infection, thus preventing their conversion to LL-FMs and the development of active tuberculosis. Furthermore, as FFMs do not respond to signals for epithelioid transformation, these macrophages could reach the caseum of granulomas, which could also explain why statins regress tuberculoid lesions in a mouse model with human-like necrotic pulmonary granulomas [90], and also accelerate *Mycobacterium tuberculosis* clearance in pulmonary TB in humans [91]. Findings in the present work may support the use of statins as adjuvant drugs in treating tuberculosis but, they also alert on the potential of these drugs to exacerbate pre-existing inflammatory diseases linked to a sterile activation of the NLRP3 inflammasome [92], for example, as observed in diabetes mellitus type I [93]. The inflammatory potential of statins could also potentiate the immune reconstitution inflammatory syndrome associated with anti-tuberculosis treatments [94]. Importantly, in patients developing post-primary tuberculosis, granulomatous structures are not formed, and the pathogen transforms numerous alveolar macrophages into LL-FMs containing mycobacterial antigens. When these cells encounter highly sensitized T-cells, a fatal massive necrotizing hypersensitivity reaction (Koch’s phenomenon) is trigged [95]. Thus, in these patients, statins could exacerbate this reaction, thereby complicating the pathology.

The possibility of modulating cholesterol metabolism to help macrophages in fighting *Mtb* provides a valuable tool for designing innovative therapies to treat tuberculosis. It even opens an unexplored possibility to generate in vitro efficient pro-inflammatory FMs that could be adoptively transferred to the hosts to help these in the control of different infectious processes. However, as the course of tuberculosis shows different clinical manifestations and outcomes according to patient’s immune status [96], such a strategy may require personalized adaptations

## 5. Conclusions

We conclude that, in the incipient GLSs, most macrophages focus on isolating the bacilli and only a few of them activate caspase-1, thereby limiting the activation of an efficient IFNγ/cytotoxic response. Fluvastatin converts macrophages into foam cells that produce NLRP3 inflammasome components in a dysregulated manner, due in part to the drug’s interference with a metabolism/immunity integrated circuit. As a result, in response to inactivated *Mtb H37Ra*, fluvastatin-treated macrophages do not form GLSs, but they assemble NLRP3 inflammasome components and markedly activate caspase-1, promoting a potent cytotoxic/IFNγ response. Therefore, although healthy donors appear to have generated a specific response against *Mtb H37Ra*, the formation of GLSs may limit its activation. In hypercholesterolemic patients, treatment with statins could improve the effectiveness of FMs to control the pathogen before granuloma starts to be formed, preventing the development of active tuberculosis. We propose that the control of *Mtb* infection requires not only the generation of specific T-cells, but also helping macrophages to activate them. The important metabolism/immunity connections described here open a series of pharmacological possibilities not only to help immunity fight tuberculosis or other infectious processes, but also to regulate some inflammatory diseases associated with alterations in cholesterol metabolism.

## Figures and Tables

**Figure 1 cells-13-00536-f001:**
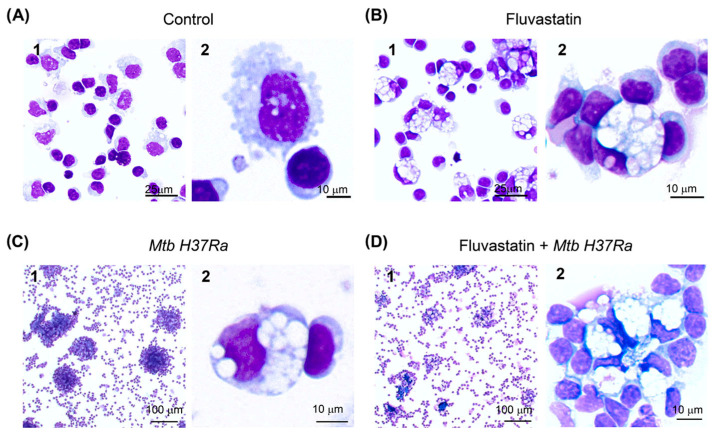
Changes undergone by macrophages of PBMCs under the different incubation conditions. (**A**) Macrophages in untreated PBMCs (control) show small vacuoles (**1**). A macrophage from this condition is shown in detail (**2**). (**B**) Fluvastatin treatment of PBMCs induces a generalized conversion of monocytes/macrophages to highly vacuolated cells, similar to conventional foamy macrophages, here referred to as FFM (**1**). Note how these macrophages tightly attach to several monocytes and/or lymphocytes (**2**). (**C**) In the absence of fluvastatin, inactivated *Mtb H37Ra* induces in PBMCs the formation of granuloma-like structures (GLSs) (**1**) and transforms a few macrophages into foamy cells similar to FFMs (**2**). (**D**) In fluvastatin-treated cultures, the exposition to *Mtb H37Ra* induces cellular aggregates rather than compact GLSs (**1**). These aggregates are mainly formed by FFMs (**2**). Cells were stained with May–Grünwald Giemsa solutions and analyzed by optical microscopy.

**Figure 2 cells-13-00536-f002:**
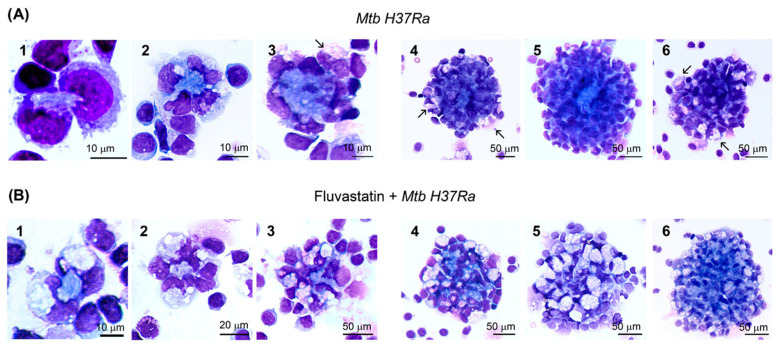
Effect of fluvastatin in the time course response of PBMCs to inactivated *Mtb H37Ra* exposition. (**A**) In cultures only exposed to the bacteria, during the first hour of incubation, two or a few macrophages initiate the formation of granuloma-like structures (GLSs) by surrounding small amorphous masses (**1**,**2**). In the following three hours, these masses progressively grow and appear encircled by some broken nuclei and by a few new recruited macrophages (**3**). Thereafter, many more macrophages enclosed these amorphous formations in compact GLSs (**4**,**5**). Note that some FMs are also incorporated (arrows in (**3**,**4**)). Many more macrophages are progressively incorporated to the structures, which continue growing until the end of the incubation (**5**). At this time, most GLSs are highly compacted, intensely stained and scarcely contain FM; however, a few GLSs are less compacted, enclose a smaller core, and incorporate more FMs (arrows in (**6**)). (**B**) Similarly, in fluvastatin-treated cultures exposed to inactivated *Mtb H37Ra*, two or more FFMs are surrounded by an amorphous mass during the first hour of incubation, but they evolve into small cores surrounded by numerous vacuolated macrophages (**1**–**3**). Later, more FFMs surround them (**4**,**5**), forming small non-compacted cell aggregates (compare (**B5**,**A5**)). Occasionally, some aggregates have large central cores and fewer FFMs (**6**). Cells were stained with May–Grünwald-Giemsa solutions and analyzed by optical microscopy.

**Figure 3 cells-13-00536-f003:**
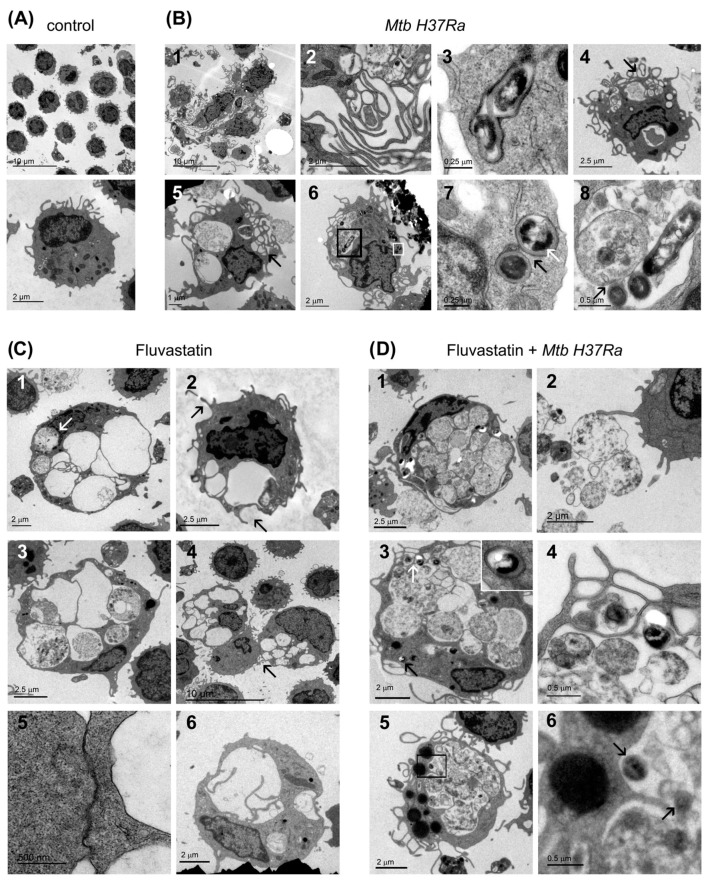
Ultrastructural characteristics of macrophages. (**A**) Resting macrophages in control PBMCs. (**B**) In inactivated *Mtb H37Ra* stimulated cultures, macrophages become epithelioid cells (**1**) that associate through a net of filopodia (**2**) and contain phagosomes that scarcely degrade bacteria (**3**). These cultures also show a few FMs with large cytoplasmic vacuoles and some smaller ones at the margin of the cell ((**4**,**5**), black arrows). Some of these FMs enclose a few bacilli within phagosomes (white square in (**6**), enlarged in (**7**)). In 7, two phagosomes are shown (black arrow), one of them containing undegraded bacteria (white arrow). The same macrophage also phagocytizes bacteria, along with debris, delivering them into the same vacuole (black square in (**6**), enlarged in (**8**)). Note that in this vacuole, three bacteria show disorganized membranes at the contact points with a vesicular organelle ((**8**), black arrow). (**C**) FFMs contain numerous vacuoles that in some cases are closely associated to the nuclei ((**1**), white arrows). These vacuolated macrophages emit phagocytic filopodia ((**2**), black arrows), and enclose vesicular organelles in their vacuoles (**3**). These cells tightly attach to a small cell through an intercellular membrane association (**4**), one of which (arrow) is enlarged in (**5**). FMs can emit filopodia-like projections to subdivide vacuoles (**6**). (**D**) FFMs exposed to inactivated *Mtb H37Ra* do not show well defined vacuoles, and their cytoplasm is filled with numerous vesicular organelles and debris, apparently trapped by a poorly defined network of filopodia (**1**). These organelles can be released out of the cells to be recognized by filopodia emitted by other cells (**2**). FMs also contain undegraded bacteria inside phagosomes ((**3**), black arrow, enlarged in the inserted image), or in vacuoles where they show signs of degradation ((**3**), white arrow, enlarged in (**4**)). Note the presence in these macrophages of large lipid bodies in the cytoplasm (**5**) and nearby, partially degraded bacteria inside vacuoles ((**5**), square and (**6**), black arrows).

**Figure 4 cells-13-00536-f004:**
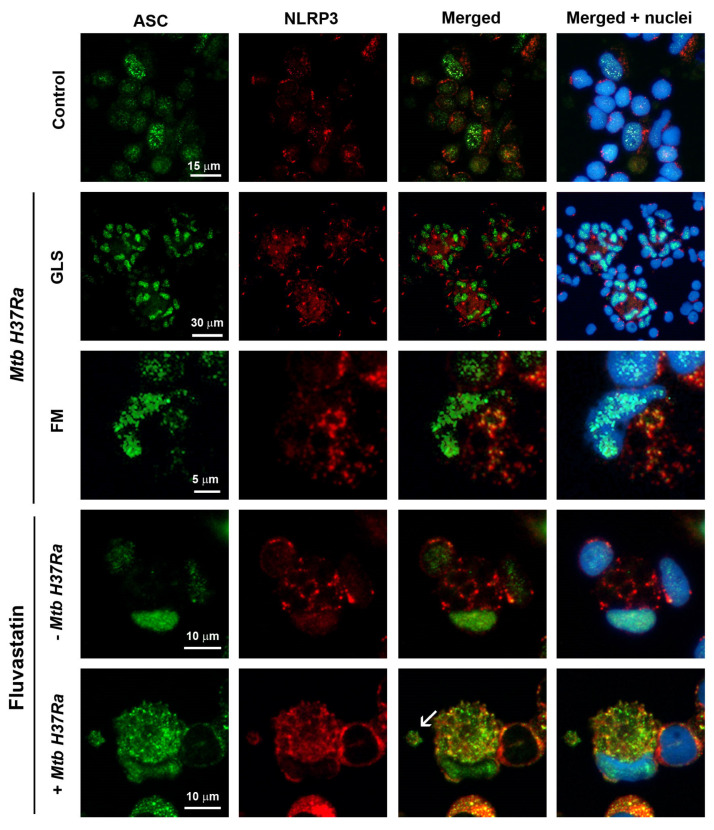
Cellular localization of ASC and NLRP3 in macrophages from different experimental conditions. Microphotographs show immunodetection of ASC (green), and NLRP3 (red), and Hoechst-stained nuclei (blue). Control row shows resting macrophages with some ASC inside the nucleus and NLRP3 forming small perinuclear dots. In *Mtb H37Ra* most cells in GLSs show increased ASC in their nuclei and concentrate NLRP3 in single perinuclear dots (merged + nuclei column). In FMs induced in response to the bacteria, NLRP3 forms a few ring-shaped aggregates in the cytoplasm. ASC increases in the nucleus, and partially moves to the cytoplasm to co-localize with NLRP3 (yellow in merged column). Under fluvastatin conditions, FFMs markedly increase ASC in the nucleus and NLRP3 in the cytoplasm where it forms numerous ring-shaped structures (merged + nuclei column). In response to inactivated *Mtb H37Ra*, FFMs show high amounts of ASC that has moved from the nucleus to the cytoplasm to co-localize with NLRP3 (yellow in merged + nuclei column). Note the presence of free extracellular particles where ASC partially co-localizes with NLRP3 (white arrow in merged column).

**Figure 5 cells-13-00536-f005:**
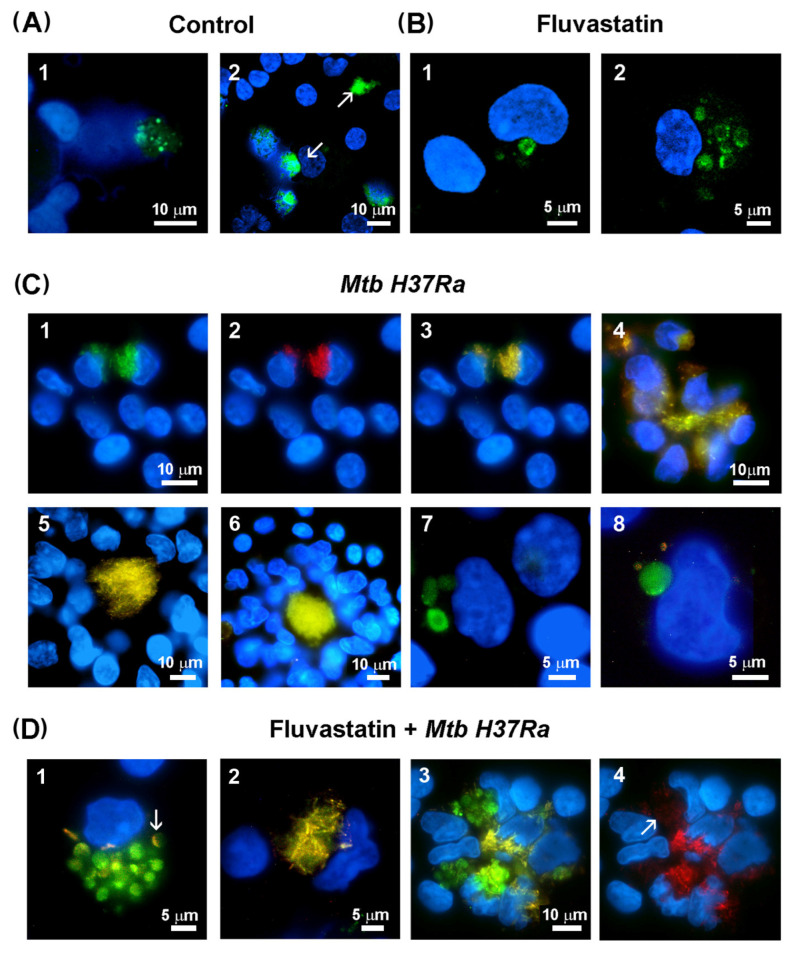
Detection of active caspase-1 in PBMCs by the FAM-FLICA assay. Green fluorescence is emitted by FAM-YVAD-FMK (specific ligand of caspase-1) and cells nuclei are stained with Hoeschst. (**A**) Control PBMCs show specific green fluorescence as dots in the broken nuclei of some macrophages (**1**,**2**) or forming large extracellular aggregates ((**2**), white arrows). (**B**) In fluvastatin-treated PBMCs, a variable number of FFMs show specific green fluorescence forming a ring near the nucleus (1) or several rings and arcs in the cytoplasm (**2**). (**C**) In PBMCs exposed to inactivated *Mtb H37Ra*, a few macrophages show large numbers of bacteria emitting green (**1**) and red (**2**) autofluorescence. Merged 1 and 2 images delineate the bacteria in yellow and evidence the absence of specific green fluorescence in these cells (**3**). Several of these macrophages associate to accumulate the bacilli at the same point (**4**). In the incipient GLS, the accumulated bacteria form a central core that is surrounded by other macrophages with a very low bacterial load, and that do not show active caspase-1 either (**5**,**6**). In this condition, only some FMs show specific green fluorescence inside a few cytoplasmic round formations. Note that these cells do not contain intact yellow bacilli (**7**,**8**). (**D**) In fluvastatin-treated cultures exposed to inactivated *Mtb H37Ra*, most FFMs show an important increase in the number of round formations, containing specific green fluorescence (1). These macrophages barely show intact bacteria, but they contain some bacilli remnants inside these round formations enclosing the active enzyme (1). In some of these formations, these remnants accumulate at their edges ((**1**), white arrow). In the same cultures, a few macrophages accumulate large numbers of bacteria without activating caspase-1 (**2**). These macrophages also accumulate bacilli in a central area ((**3**), yellow) and some bacteria locate into broken nuclei. Near them, other FFM contain numerous round formations filled with active caspase-1 ((**3**), green) unmasking numerous red auto-fluorescent small particles indicating bacterial degradation ((**4**), white arrow).

**Figure 7 cells-13-00536-f007:**
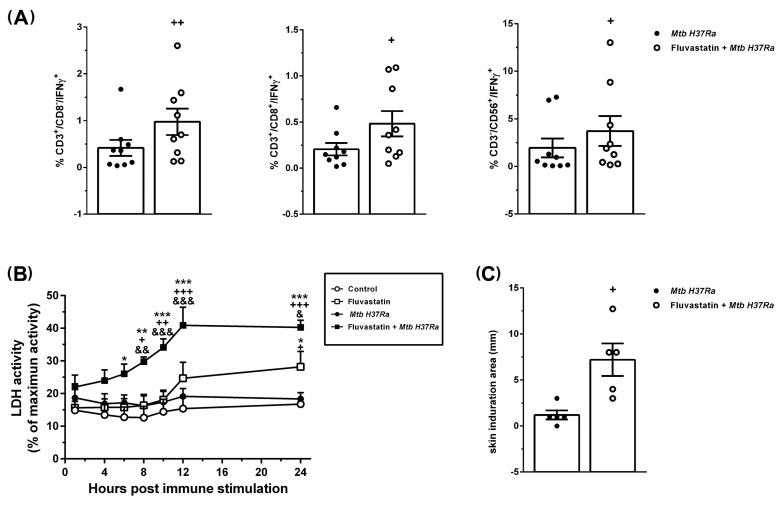
Fluvastatin exacerbates the cellular response against *Mtb H37Ra*. (**A**) Percentage of CD3^+^CD8^−^, CD3^+^CD8^+^ and CD3^−^CD56^+^ cells that produce IFNγ in untreated (black circles) or fluvastatin-treated PBMCs exposed to *Mtb H37Ra* (open circles). (**B**) Time-course of LDH activity in culture media. Note the significant increase in cytotoxicity promoted by fluvastatin in cells exposed to *Mtb H37Ra*. (**C**) Response to tuberculin test in rabbits immunized with *Mtb H37Ra* that previously received fluvastatin (open circles) or vehicle (black circles). Results in (**A**,**B*)*** represent the mean ± SE of 9 (**A**), or 5 (**B**) independent experiments. * *p* ≤ 0.05, ** *p* ≤ 0.01, *** *p* ≤ 0.001 vs. control; ^+^ *p* ≤ 0.05, ^++^ *p* ≤ 0.01, ^+++^ *p* ≤ 0.001 vs. *Mtb H37Ra*; ^&^ *p* ≤ 0.05, ^&&^ *p* ≤ 0.01, ^&&&^ *p* ≤ 0.001 vs. fluvastatin (Two-way ANOVA followed by Neuman–Keuls test). In (**C**), results represent the mean ± SEM of 5 rabbits for each experimental condition. ^+^ *p* ≤ 0.05, ^++^ *p* ≤ 0.01 vs. *Mtb H37Ra* (paired *t*-test).

## Data Availability

The datasets generated and/or analyzed in the current study are available from the corresponding author on reasonable request.

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
