# Peer review of "Fluvastatin Converts Human Macrophages into Foam Cells with Increased Inflammatory Response to Inactivated Mycobacterium tuberculosis H37Ra"

_cells, 2024, doi:10.3390/cells13060536_

Round 1

Reviewer 1 Report (Previous Reviewer 1)

Comments and Suggestions for Authors

Some of my concerns have been addressed but not all. Specifically:

1. While the Discussion has been shortened there is still far too much conjecture that is presented as if it was factual. I strongly suggest the authors revise the Discussion with reference to the data presented in figures 1 - 5 and clearly segregate what they have actually observed (and can show in the images) from their interpretation of what it might mean and consider where there may be alternative interpretations. It is OK to hypothesise about what the results might mean but it should be clear that this is the case by using qualifiers such as may/might/could rather than is/would. It detracts from otherwise interesting findings when purely qualitative results (in Figures 1-5) are over interpreted in this way. 

It is also important to acknowledge that the granulomas are made with killed attenuated Mtb. So, while the model has its merits within certain limits, it is important not to extrapolate too much from the results on the possible effects of live virulent Mtb on inflammasome activation which may be much different. The main focus of the manuscript should be the novel findings with Fluvastatin.

Examples of over interpretation include but are not confined to lines 779 – 803. The authors show no evidence that NLRP3 is directly interacting with its ligands so why bring this up.

Line 684 – Is this sentence referring to H37Ra-stimulated cells? At 24h the IL-1b level is around 3000pg/ml so that’s not a “poor” level, just less than that produced in the presence of the statin. It would be more correct to state that lower levels of IL-1b were produced compared to cells treated with the statin plus Mtb.

Delete the first sentence of the Conclusions (line 865- ). Again, this is over interpretation of results obtained with dead bacteria.

2. On the question of whether fluvastatin influences uptake of Mtb bacilli by macrophages: the EM results in figure 3 are not quantitative and are open to interpretation as to whether the difference is due to uptake or degradation of phagocytosed bacilli. It would be easy to incubate PBMCs with H37Ra +/- fluvastatin for a short period of time (e.g. 2-4h), followed by acid fast staining or another method to visualise Mtb, to see if the drug has any effect on uptake. Alternatively, the fact that this was not done could be acknowledged as a limitation of the study. 

3. With regards to my comments on whether the cells are undergoing necrosis or not, I take the authors point about LDH measurement which suggests that necrosis is occurring with the combination of fluvastatin plus H37Ra and to a lesser extent with fluvastatin alone. However, there is no evidence that killed Mtb is causing necrosis in this model. Most of the images that purport to show necrosis do not do so in my opinion.

    • Fig 1C1 – is too low power to see a necrotic core.
    • Fig 2A1 and 2 - I see no evidence of a necrotic core in 2b 1 and 2.
    • In the case of fluvastatin + Ra the nuclei may have become misshapen due to necrosis or are compressed by the large swollen vacuoles taking up most of the cytoplasm

Consequently, they need to correct this in the results section or, if they are convinced the killed H37Ra is causing significant amounts of cell death, provide more credible evidence for this. Similarly, in the Discussion refrain from speculating too much about this topic with regard to Mtb alone. Cell death may be a factor in fluvastatin-treated cells responses.

4. Line 781-782 - There may not have been much caspase 1 activity detected but fluvastatin alone was able to stimulate the secretion of large amounts of IL-1b and IL-18 even if it was increased in the presence of the immune stimulus.

Minor points:

5. Figure 2 - include the times that images were taken in the legend.

6. Line 647 – this should read “blood-derived macrophages” instead of “blood macrophages”.

Comments on the Quality of English Language

The English is good. Minor editing required.

Author Response

  1. While the Discussion has been shortened there is still far too much conjecture that is presented as if it was factual. I strongly suggest the authors revise the Discussion with reference to the data presented in figures 1 - 5 and clearly segregate what they have actually observed (and can show in the images) from their interpretation of what it might mean and consider where there may be alternative interpretations. It is OK to hypothesise about what the results might mean but it should be clear that this is the case by using qualifiers such as may/might/could rather than is/would. It detracts from otherwise interesting findings when purely qualitative results (in Figures 1-5) are over interpreted in this way. 

Trying to solve the reviewer’s concerns we have read with much attention the discussion of the manuscript that was resubmitted to Cells (cells_2858743). Following the reviewer’s suggestions we have modified some sentences of the discussion using qualifiers as “could”, etc., and we have included in some sentences the verb “to suggest” to differentiate possible interpretations of the results from what we clearly observe. All these changes are highlighted in read in the new version of this resubmitted manuscript.

It is also important to acknowledge that the granulomas are made with killed attenuated Mtb. So, while the model has its merits within certain limits, it is important not to extrapolate too much from the results on the possible effects of live virulent Mtb on inflammasome activation which may be much different.

The main focus of the manuscript should be the novel findings with Fluvastatin.

We selected Mtb H37Ra as a source of Mtb PAMPs because this strain lacks virulence factors and shares most of their membrane proteins with their virulent counterpart Mtb H37Rv, what confer them the capacity to elicit a similar profile of cytokines [35,36]. See lines 115-123 of materials and methods.

Delete the first sentence of the Conclusions (line 865- ). Again, this is over interpretation of results obtained with dead bacteria

In the Conclusions we have used by mistake the term granuloma rather GLS several times.  By doing this, our conclusions sound as if we extrapolated our results to what happens in true granulomas. As this was not our purpose, we apologize for it. We have corrected them and rewritten the first sentence of the introduction as follows:

We conclude that, in the incipient GLS, most macrophages focus on isolating the bacilli and only a few of them activate caspase-1, thereby limiting the activation of an efficient IFNg/cytotoxic response.

Examples of over interpretation include but are not confined to lines 779 – 803. The authors show no evidence that NLRP3 is directly interacting with its ligands so why bring this up.

As mentioned above, we have read with much attention the discussion of the resubmitted manuscript to Cells (cells_2858743) and found that nothing in this fragment is discussed about the way in which Mtb PAMPs promote NLRP3 receptor assemblage. Perhaps, it may have been an error in line numbers, and the reviewer refers to lines 768-778 of the formerly submitted manuscript. In order to exclude any overinterpretation of our results, we have modified this paragraph as follows:

It has been described that different cellular organelles, which vary between different cells, can act as supports for the trafficking and assembly of the inflammasome components, thus mediating different responses [84]. In our study, FFMs respond to Mtb by assembling the NLRP3 inflammasome and activating caspase-1 around numerous organelles that degrade the bacteria and that retain the enzyme once it has been activated. The presence of bacilli remnants forming arcs at the edges of these organelles suggests that bacilli degradation may provide inflammasomes with PAMPs to generate a signal 2 that fully activates caspase-1, a possibility not described so far [85].

Line 684 – Is this sentence referring to H37Ra-stimulated cells? At 24h the IL- 1b level is around 3000pg/ml so that’s not a “poor” level, just less than that produced in the presence of the statin. It would be more correct to state that lower levels of IL-1b were produced compared to cells treated with the statin plus Mtb.

As suggested by the reviewer, we have slightly modified previous line 684 (now 698-699) as follows: 

Consequently, the secretion of IL-1b and Il-18 was not significantly different from that observed in control cultures.

  1. On the question of whether fluvastatin influences uptake of Mtb bacilli by macrophages: the EM results in figure 3 are not quantitative and are open to interpretation as to whether the difference is due to uptake or degradation of phagocytosed bacilli. It would be easy to incubate PBMCs with H37Ra +/- fluvastatin for a short period of time (e.g. 2-4h), followed by acid fast staining or another method to visualize Mtb, to see if the drug has any effect on uptake.

Our EM results mainly show that fluvastatin change macrophages morphology and functional polarization in response to the bacteria. Although these macrophages enclose undegraded bacteria, we have not conducted a rigorous study that would allow us to determine whether these macrophages phagocytose and/or degrade the bacterium, so we are agree with the reviewer that the EM results are open to interpretation. Although we have not performed an acid-fast staining, we used Mtb autofluorescence as a tool to visualize quantitative differences in the uptake of the bacteria by macrophages from the different conditions (see lines 208-218 in material and methods). Thus, results presented in Fig. 5D and Supplementary Fig. 3 suggest that both, fluvastatin treated macrophages and FM induced by the bacteria only, are able to uptake and degrade the bacilli, whereas the remaining macrophages do not do it. 

  1. With regards to my comments on whether the cells are undergoing necrosis or not, I take the authors point about LDH measurement which suggests that necrosis is occurring with the combination of fluvastatin plus H37Ra and to a lesser extent with fluvastatin alone. However, there is no evidence that killed Mtb is causing necrosis in this model. Most of the images that purport to show necrosis do not do so in my opinion. Consequently, they need to correct this in the results section or, if they are convinced the killed H37Ra is causing significant amounts of cell death, provide more credible evidence for this. Similarly, in the Discussion refrain from speculating too much about this topic with regard to Mtb alone. Cell death may be a factor in fluvastatin-treated cells responses.

Fig 1C1 – is too low power to see a necrotic core.

We agree with you and we have changed in line 323 “necrotic cores” by “central cores”

Fig 2A1 and 2 - I see no evidence of a necrotic core in 2b 1 and 2.

You are right, in lines 348-361 we have referred images from figure 2 in a wrong place.

We have rewritten this paragraph, also indicating periods of time at which images were observed, as follows:

During the first hour of immune stimulation, we observed a small number of formations in which two or a few more macrophages surrounded small amorphous masses (Figure 2A 1 and 2). A little later, these formations evolved toward larger amorphous masses surrounded by dismantled nuclei from the initial macrophages and a few other macrophages, including FM, which arrived later (figure 2A 3). This result indicates that some of the macrophages initiating the formation died by necrosis and debris generated got mixed with the accumulated material. Thus, we will refer to them as necrotic cores or necrotic areas. We could not determine the number of macrophages that died in this action, but it seemed to be very small. In the following hours a high number macrophages quickly incorporated to these formations enclosing them in the core of small GLS (figure 2A 4). During the rest of the incubation period, many more macrophages were incorporated to GLS, which increased in size. These newly recruited macrophages became compacted over time and the structures stained more intensely in blue. As observed in figure 2A 5 the central core also grew over time.

We have also corrected lines 368-370 in this section as follows:

In cultures treated with fluvastatin and exposed to the bacilli, a few FFM also surrounded similar amorphous masses (Figure 2B 1 and 2) but they barely grew over time and were surrounded by more FFM.

As you comment, regarding results presented in figure 1, 2 and the cytotoxicity topic in the text, we cannot say that, in our model, killed Mtb causes macrophage necrosis. In Figure 5C, we show evidence that the early amorphous formations referred above are actually large amounts of bacteria surrounded by a few macrophages, some of which die by necrosis. Later, they are isolated by other living macrophages that scarcely contain bacilli. We propose that there is a small subset of macrophages capable of capturing so large quantities of bacteria that they die by necrosis. A few of these macrophages associate to form necrotic nuclei filled with the captured bacilli. We thought that the number of these necrotizing macrophages is really low and that the cytotoxicity test does not have enough sensitivity as to differentiate small increases of LDH.

In the case of fluvastatin + Ra the nuclei may have become misshapen due to necrosis or are compressed by the large swollen vacuoles taking up most of the cytoplasm

Indeed, both of the reviewer's proposals are feasible. Alternatively, by observing Fig. 5D, 4, we could also propose that macrophages accumulating bacteria in some way concentrate them into their nuclei to initiate a necrotic process.

  1. Line 781-782 - There may not have been much caspase 1 activity detected but fluvastatin alone was able to stimulate the secretion of large amounts of IL-1b and IL-18 even if it was increased in the presence of the immune stimulus.

As shown in Fig. 6A, neither Mtb nor fluvastatin significantly affected caspase-1 activity at any time analyzed, as compared with control PMBCs. Similar results were observed for IL-18. Although it seems that fluvastatin stimulates IL-1b release, the statistical analysis (see the methods section) does not show any significant difference when compare all the experimental conditions and times of study. Nevertheless, and trying to explain this trend of IL-1β but the lack of significance, we have introduced the following sentences in lines 770-771:

In agreement with this, there was an increasing trend in caspase-1 activity in cellular lysates from this condition as compared to control cells.

Minor points:

  1. Figure 2 - include the times that images were taken in the legend.

The times at what images were seen have been included in the legend.

  1. Line 647 – this should read “blood-derived macrophages” instead of “blood macrophages”.

We have changed it.

The English is good. Minor editing required.

We have performed the minor English revision requested by referee.

Reviewer 2 Report (New Reviewer)

Comments and Suggestions for Authors

The paper titled "Fluvastatin converts human macrophages into pro-inflammatory foam cells with therapeutic potential in tuberculosis" by Montero-Vega and colleagues provides a nuanced perspective on the role of Fluvastatin in inflammation. The authors' arguments are supported by extensive experimental evidence, and their data interpretation is technically very sound. This paper is suitable for publication in Cells, but the reviewer requests some minor revisions:

Minor issues:

1. The Introduction section, particularly from Page 2, line 61 to line 93, is overly long. It is suggested to streamline the discussion by focusing on HMG-CoA and trimming down the sections related to PAMP and DAMP immune responses.

2. The Methods section is generally well-described. However, more details are needed for the 2.5 FAM-FLICA assay. Clarifications on concentrations, experimental duration, and procedural steps would be beneficial.

3. The figures in the Supplementary Materials contain valuable data that could be moved to the main manuscript as regular figures. Consider relocating them for better visibility and accessibility.

Author Response

  1. The Introduction section, particularly from Page 2, line 61 to line 93, is overly long. It is suggested to streamline the discussion by focusing on HMG-CoA and trimming down the sections related to PAMP and DAMP immune responses.

As suggested by the reviewer we have reduced lines 77 to 89 of the introduction as follows:

Inflammasomes assembly requires two cell signals. Signal 1 is generated by membrane pattern recognition receptors (PRRs) after sensing extracellular pathogen associated molecular patterns (PAMPs), and/or danger associated molecular patterns (DAMPs) and induces up-regulation of the inflammasome components, pro-caspase-1 and pro-IL-1b.  Signal 2 is mediated by PAMPs, DAMPs, cellular stressors and even metabolic perturbations and induce conformational rearrangement of a specific cytosolic PRRs, thereby initiating its assemblage. Most inflammasomes oligomerize and assemble with ASC (apoptosis-associated speck-like protein containing a CARD), an adaptor protein that recruits pro-caspase-1 to the structure for their autoprocessing and activation.

  1. 2. The Methods section is generally well-described. However, more details are needed for the 2.5 FAM-FLICA assay. Clarifications on concentrations, experimental duration, and procedural steps would be beneficial.

You are right. In the manuscript, we have now specified that we isolated PBMCs and incubated them for a period of 15h plus 24h under identical conditions to those described in section 2.2. The FLICA reagent is supplied as a lyophilized powder that must be dissolved (see below), but the amount of product in each vial is not specified. Thus, we cannot calculate the precise concentrations used. We also explain how we dissolved this reagent to prepare a 30X working solution used in the assay, to avoid the interference of DMSO with caspase-1 activation, (see lines 189-196).

3.The figures in the Supplementary Materials contain valuable data that could be moved to the main manuscript as regular figures. Consider relocating them for better visibility and accessibility.

Thank you very much for this consideration. We agree that it could be useful, however, as the manuscript already include 8 figures, adding three more could be excessive. We thought that there is not difficult to access them.

Reviewer 3 Report (New Reviewer)

Comments and Suggestions for Authors

This study investigated in a methodologically sound way the very important issue of the effect of statins on macrophages. The manuscript can be improved by mentioning in the introduction relevant content of the following publication:

Sun HY, Singh N. Potential role of statins for the management of immune reconstitution syndrome. Med Hypotheses. 2011 Mar;76(3):307-10.

In the discussion section the authors need to discuss the potential implications of their findings for investigation of the presentation of tuberculosis in patients on statin treatment including the result of tuberculin skin tests for diagnosis of M. tuberculosis infection and the treatment induced immune reconstitution inflammatory syndrome which occurs in about 10% of patients on treatment for active tuberculosis (TB-IRIS). The need to mention the potential increased risk of dissemination of tuberculosis on this treatment and how to investigate this by checking the association of miliary tuberculosis with statin treatment.

Author Response

  1. This study investigated in a methodologically sound way the very important issue of the effect of statins on macrophages. The manuscript can be improved by mentioning in the introduction relevant content of the following publication:

Sun HY, Singh N. Potential role of statins for the management of immune reconstitution

We have included in the introduction the following sentence:

In addition, other authors have hypothesized a potential beneficial effect of statins in the management of the of immune reconstitution syndrome [20] (lines 69-71).

  1. In the discussion section the authors need to discuss the potential implications of their findings for investigation of the presentation of tuberculosis in patients on statin treatment including the result of tuberculin skin tests for diagnosis of M. tuberculosis infection and the treatment induced immune reconstitution inflammatory syndrome which occurs in about 10% of patients on treatment for active tuberculosis (TB-IRIS). The need to mention the potential increased risk of dissemination of tuberculosis on this treatment and how to investigate this by checking the association of miliary tuberculosis with statin treatment.

Following your indications, we have added in the new version of the manuscript the following sentences:

- Lines 815-819:

Since FFM cannot form GLS, statins could promote the dissemination of the bacilli. Yet, this possibility seems unlikely because the drug does not prevent macrophages from accumulating the bacilli in central areas, which are then surrounded by numerous FFM that effectively capture and destroy the accumulated bacteria. Based on our results,

- lines 828-830.

This finding raises the possibility that the tuberculin test may generate false positives or be exacerbated in those patients treated with statins.

- In addition, lines 842-856 have been rewritten as follows:

The inflammatory potential of statins could potentiate the immune reconstitution inflammatory syndrome associated to anti-tuberculosis treatments [94]. Importantly, in patients developing post-primary tuberculosis, granulomatous structures are not formed, and the pathogen transforms numerous alveolar macrophages into LL-FM containing mycobacterial antigens. When these cells encounter highly sensitized T cells, a fatal massive necrotizing hypersensitivity reaction (Koch's phenomenon) is trigged [95]. Thus, in these patients, statins could exacerbate this reaction, thereby complicating the pathology.

The possibility of modulating cholesterol metabolism to help macrophages in fighting Mtb provides a valuable tool for designing innovative therapies to treat tuberculosis. It even opens an unexplored possibility to generate in vitro efficient pro-inflammatory FM that could be adoptively transferred to the hosts to help these in the control of different infectious processes. However, as the course of tuberculosis shows different clinical manifestations and outcomes according to patient’s immune status [96], such a strategy may require personalized adaptations.

This manuscript is a resubmission of an earlier submission. The following is a list of the peer review reports and author responses from that submission.

Round 1

Reviewer 1 Report

Comments and Suggestions for Authors

This manuscript by Montero-Vega et al investigates the effects of fluvastatin on human monoyctes. They use in vitro TB granuloma generated from PBMCs and heat-treated H37Ra to study the effects of fluvastatin on monocytes/macrophages in the presence of M. tuberculosis (Mtb). These cells form granulomas with a necrotic core surrounded by epitheloid cells containing mycobacteria. In contrast monocytes in cultures of PBMCs that were exposed to fluvastatin formed looser aggregates of cells with absent or smaller necrotic cores and no epitheloid cells showed less evidence of intact intracellular mycobacteria. Fluvastatin treatment also amplified the inflammatory response to H37Ra in terms of proinflammatory cytokine secretion via activation of the inflammasome. Fluvastatin treatment in the absence of H37Ra induced a vacuolated phenotype in monocytes/macrophages.

Overall, this is an interesting study, and the in vitro generated granuloma model is a useful model for studying foam cell biogenesis and function. However, I have some concerns with their interpretation of the data.

Specific comments:

1. One of the main questions I have is whether the vacuolated macrophages resulting from fluvastin treatment can be accurately described as foamy macrophages. The term foamy macrophages typically is used to describe lipid-laden cells whereas the vacuoles in the fluvastatin treated cells do not appear to contain lipids. If this type of vacuolated cell has previously been described as FM please provide supporting evidence.

2.  Some statins can inhibit phagocytosis. Is this the case with Fluvastatin? Has the uptake of H37Ra been assessed in the presence and absence of fluvastatin treatment? If fluvastatin pre-treatment inhibited uptake of H37Ra this could be the reason that there are less bacteria in the macrophages.

3.  This is a descriptive study and the authors should be careful not to over interpret the results. For example, there is a long speculative discussion (approx. 2 pages) about the H37Ra-induced granuloma. This seems excessive given that the paper is about the effects of fluvastatin. It would be better to focus on that in the discussion or at least get to it sooner.

4. Lines 644-647 “suggesting that these macrophages undergoing necrosis could emit some unknown signal that initiate the formation of granulomas.”

Line 678 -its not clear that the macrophages are turning into FM due to "necrotic debris"

805-7 - FFM do not respond to necrotic signals for epithelioid transformation, and thereby they do not form GLS ..

Published evidence indicates that necrosis is not necessary for granuloma formation. It seems more likely that the granulomas are formed as a response to Mtb antigen or antigens. In addition, cell death was not measured so it is speculation that the cells underwent necrosis.

5. Line 838 “first encounter with Mtb… “Previous exposure to M. tuberculosis or BCG would influence the production of IFN gamma by T cells. Were the buffy coat donors tested for PPD reactivity or by IGRA? If not it is impossible to know if this was a first encounter of their cells with Mtb.

6. Line 684 "FM cannot convert macrophages to LL-FM" - what does this mean? That FM cannot be converted to LL-FM due to a lack of live Mtb? How can you tell that he vacuolated macrophages would progress to LL-FM if live bugs were present?

Minor comments:

Delete lines 40-47.

There are quite a few non-standard abbreviations – a list of abbreviations would be useful for the reader.

Figure 3 and 4 Arrows in photomicrographs could be bigger – they are difficult to see.

Line 641 “a not described strategy”.  It could better be changed to “an undescribed strategy” in this context.

“undescribed” is used a lot in the manuscript. Previously undescribed would sound better.

Comments on the Quality of English Language

Minor editing is required.

Author Response

Responses to the comments of referee Number 1:

  1. One of the main questions I have is whether the vacuolated macrophages resulting from fluvastin treatment can be accurately described as foamy macrophages. The term foamy macrophages typically is used to describe lipid-laden cells whereas the vacuoles in the fluvastatin treated cells do not appear to contain lipids. If this type of vacuolated cell has previously been described as FM please provide supporting evidence.

Nowadays, the term “foamy (or foam) cell” has become widespread and is used by most immunologists to refer to lipid-laden macrophages alike those present in atherosclerosis lesions (reference 61 in the manuscript). However, pathologists consider these cells as histiocytes that help in healing the body after an injury or infection by removing dead cells, blood, micro-organisms (such as bacteria and fungus), and foreign material (reference 46 in the manuscript). As these cells are mostly macrophages the term foamy histiocyte has given rise to foamy macrophage. As mentioned in the discussion (lines 656-673), the biogenesis and function of FM vary depending on the specific microenvironment of each disease, where these cells can accumulate not only lipids but also other products derived from the waste engulfed from the damaged tissue, helping pathologists in the diagnosis. It is considered that, in the incipient granulomas, foamy macrophages are recruited to remove debris generated during the early immune response to Mtb, but they become infected and convert to lipid-laden FM by the bacteria (lines 45-47 in the introduction). For a better illustration of the similarity between the different highly vacuolated macrophages generated in our work and conventional FM, in figure 1S we compare them with erythrophages, which are known by pathologists as FM that clean erythrocytes in hemorrhagic tissues to recover their homeostasis.

  1. Some statins can inhibit phagocytosis. Is this the case with Fluvastatin? Has the uptake of H37Ra been assessed in the presence and absence of fluvastatin treatment? If fluvastatin pre-treatment inhibited uptake of H37Ra this could be the reason that there are less bacteria in the macrophages.

In our present work fluvastatin does not inhibit phagocytosis itself. In figure 2C 2, we show how FM induced by fluvastatin (FFM) perform phagocytosis to engulf some undetermined material from the surrounding microenvironment. In Figures 2D 3,4 can be seen how FFM exposed to Mtb enclose some bacteria in phagosomes and phagocytose others along with debris through the same phagocytic processes as FM showed in Figure 2B 3, 6-8. These FFM, deliver the bacteria into a cytoplasm filled with organelles alike secondary lysosomes. In agreement, in Figure 5 can be seen autofluorescent bacilli remnants inside these organelles. Therefore, the reason for less bacteria in FFM would be that these organelles are able to degrade them.

  1. This is a descriptive study and the authors should be careful not to over interpret the results. For example, there is a long speculative discussion (approx. 2 pages) about the H37Ra-induced granuloma. This seems excessive given that the paper is about the effects of fluvastatin. It would be better to focus on that in the discussion or at least get to it sooner.

To analyze the effects of fluvastatin on NLRP3/caspase-1 activation, we first studied how healthy PBMCs activate this inflammasome in response to Mtb PAMPs, using inactivated Mtb H37Ra. Unexpectedly, these dead and avirulent bacteria induce the formation of epithelioid granulomas following a gradual process that mimics the first stages of the formation of these granulomas in vivo, providing valuable new information on how this process occurs. For this reason, we devoted a part of the first page of discussion to argue on the existence of previously undescribed macrophages that along bona fide epithelioid cells initiate theformation of granulomas in PBMC. Thus, we propose that our experimental model offers many advantages to design complementary studies directed to decipher the initial steps in granulomas formation. We also discuss how NLRP3 inflammasome and caspase-1 are activated by the different macrophages forming the granulomatous structure, to discuss subsequently about the effects exerted by fluvastatin. However, to avoid speculative sentences and reduce the text we have rewritten this first part of the discussion (see lines 629-673).

  1. Lines 644-647 “suggesting that these macrophages undergoing necrosis could emit some unknown signal that initiate the formation of granulomas.”

This sentence has been deleted

Line 678 -its not clear that the macrophages are turning into FM due to “necrotic debris”

According to pathologists, macrophages become foam cells in response to undescribed signals emitted by damaged tissues and penetrate these to remove debris generated to restore homeostasis. Considering this, in our conditions FM would respond to warning signals generated by necrotic cells. Therefore, we say: in our in vitro study, some macrophages become FM in response to necrotic debris generated during the earliest immune response to Mtb. However, to avoid confusion for future readers, in this sentence we have changed “necrotic debris” by “immune debris”.  

805-7 - FFM do not respond to necrotic signals for epithelioid transformation, and thereby they do not form GLS.

In this sentence, we have deleted “necrotic”.

Published evidence indicates that necrosis is not necessary for granuloma formation. It seems more likely that the granulomas are formed as a response to Mtb antigen or antigens. In addition, cell death was not measured so it is speculation that the cells underwent necrosis.

In some experiments, necrosis was quantified. Figure 7 B shows the time course of LDH activity in culture media under different incubation conditions. Although we did not find significant changes between control PBMC and PBMC exposed to the bacteria, we think that since the number of macrophages that die accumulating the bacteria is really low, the increase in LDH activity produced by them can barely be detected with this method. However, in Figure 2A 3, many nuclei initially surround a blue-stained amorphous mass, whereas in most evolved granulomas these nuclei appear broken and dismantled and cells cytoplasm’s merged with this amorphous mass. In addition, in Figure 5D 4 these macrophages showed large amounts of bacteria placed into their broken nuclei. Therefore, we have considered that this type of cell death must be necrosis and cannot be considered apoptosis or autophagy.

  1. Line 838 “first encounter with Mtb… “Previous exposure to M. tuberculosis or BCG would influence the production of IFN gamma by T cells. Were the buffy coat donors tested for PPD reactivity or by IGRA? If not it is impossible to know if this was a first encounter of their cells with Mtb.

Thank you for this comment. We have changed “first encounter with Mtb” by “during earliest stages of Mtb infection”

  1. Line 684 “FM cannot convert macrophages to LL-FM” - what does this mean? That FM cannot be converted to LL-FM due to a lack of live Mtb? How can you tell that he vacuolated macrophages would progress to LL-FM if live bugs were present?

Thanks for highlighting this confusing sentence. We want to say that FM cannot be converted to LL-FM due to a lack of live Mtb. Thus, it has been changed in the corrected manuscript (671-672).  Besides, several studies have demonstrated that Mtb virulent strains manipulate lipids metabolism in their host cell converting them into lipid load-FM (see references 4-6).

Minor comments:

Delete lines 40-47.

These lines have been deleted Lines 40-47 have been deleted in the corrected manuscript as well as boilerplate words in the title.

There are quite a few non-standard abbreviations – a list of abbreviations would be useful for the reader.

We have followed authors instructions from the Journal. In this regard they say:

Acronyms/Abbreviations/Initialisms should be defined the first time they appear in each of three sections: the abstract; the main text; the first figure or table. When defined for the first time, the acronym/abbreviation/initialism should be added in parentheses after the written-out form.

Figure 3 and 4 arrows in photomicrographs could be bigger – they are difficult to see.

As suggested by the reviewer, the arrows in these figures have been enlarged.

Line 641 “a not described strategy”.  It could better be changed to “an undescribed strategy” in this context.

As suggested by the reviewer we have changed it.

“undescribed” is used a lot in the manuscript. Previously undescribed would sound better.

Thanks a lot for this suggestion. Along the manuscript we have change “undescribed” by Previously undescribed.

Some editing has been performed throughout the article.

Reviewer 2 Report

Comments and Suggestions for Authors

 Introduction

The first part of the introduction looks like it was lifted from the ‘Instructions for the authors’. It should be removed.

Results

Sections 3.1-3.4 The results here are almost all descriptive, with little quantitation. There has to be some sort of measurement and quantitative comparison using a recognized method (eg blinded scoring) of the microscopy results described.

Discussion

The discussion is much too long. It should be reduced by a third at least.

Author Response

Response  to the comments of referee Number 2:

The first part of the introduction looks like it was lifted from the ‘Instructions for the authors’. It should be removed.

Thanks a lot for your kind indications. Lines 40-47 have been deleted as well as boilerplate words in the title.

Results

Sections 3.1-3.4 The results here are almost all descriptive, with little quantitation. There has to be some sort of measurement and quantitative comparison using a recognized method (eg blinded scoring) of the microscopy results described.

In this work we cannot evaluate the number of cells in section section 3.1-3.4 because, as observed by flow cytometry, when PBMC are exposed to Mtb practically all the macrophages in the cultures are recruited to compact granulomas or form cellular aggregates. The same is applicable to following sections. On the other hand, the main objective of this work has been to evaluate the changes induced by fluvastatin on the immune response that healthy PBMC activate against Mtb, mainly focused on the activation of the NLRP3/caspase-1 inflammasome. In this manuscript we show that fluvastatin induces a deregulated production of ASC and NLRP3. Furthermore, results in Figure 4 allow us to indicate that when combining fluvastatin treatment with the bacteria, a translocation of ASC from the nucleus to the cytoplasm occurs, to colocalize with NLRP3. These results have been obtained consistently in many independent experiments and they are in consonance with the way that caspase-1 is activated under different conditions (Figure 5). Thus, to solve the impossibility of perform a quantitative evaluation of the number of cells showing this overproduction, we assessed the final effect of fluvastatin on the activation of NLRP3/caspase-1 by quantifying caspase-1 activity in lysates of cells collected from each condition at different incubation times (Figure 6). Furthermore, we have quantified differences in the production of IL-1β, IL-18 and other cytokines, in order to evaluate the magnitude of the immune changes induced by the drug.

Discussion

The discussion is much too long. It should be reduced by a third at least.

Discussion is long due to the novelty and complexity of the topics treated there; however, its extension has been reduced. We discuss the effects of fluvastatin on NLRP3/caspase-1 activation to firstly study how healthy PBMCs activates this inflammasome in response to Mtb PAMPs, using inactivated Mtb H37Ra. Unexpectedly, these dead and avirulent bacteria induce the formation of epithelioid granulomas following a gradual process that mimics the first stages of the formation of these granulomas in vivo, providing valuable new information on how this process occurs. Considering that the Special Issue: Tuberculosis: from pathogenesis to targeted therapies, could be consulted by experts in tuberculosis form this area of research, we devoted some space to these topics and propose that our experimental model offers many advantages to design complementary studies directed to decipher the initial steps in granulomas formation. However, to avoid speculative sentences and reduce the extension of the text we have rewritten this first part of the discussion.

Reviewer 3 Report

Comments and Suggestions for Authors

M. Teresa Montero-Vega et al’s manuscript “Fluvastatin converts human macrophages into pro-inflam- matory foam cells with therapeutic potential in tuberculosis” presents that the treatment of Fluvastatin can polarize the monocytes/macrophages to foamy cells that overproduce NLRP3 inflammasome components, which then activates caspase-1 and stimulates a potent IFNγ cytotoxic response in PBMCs. It explores the mechanism of cholesterol metabolism and immunity connections and offers a potential therapy to fight tuberculosis.

Overall, this manuscript presents valuable insights into the interplay between cholesterol metabolism and the immune response in tuberculosis. It would be of interest to the existing body of knowledge in this area of research. However, there are some questions to be addressed.

1.     The scale bars in figures 4 & 5 are unclear. These scale bars need to be carefully reviewed and aligned accurately. For instance, the scale bar in Fig 5C-4 doesn't seem like 5um.

2.     In Figure 5, when both the autofluorescence and specific signals were displayed in the green channel, it was not convincing enough to distinguish between the auto-fluorescent and specific signals as the authors described. The authors should use another channel to detect caspase-1. Additionally, instead of explaining in the figure legends, proper labeling of staining and channels directly on the pictures is necessary.

3.     In this manuscript, the authors observed that Fluvastatin increased the production of inflammatory cytokines (Fig 6A, IL-1β & IL-18). However, several other studies (PLoS One.2011;6(8):e22655; Acta Cardiol. 2010 Jun;65(3):285-9.; Hepatol Res. 2013 Jul;43(7):775-84) have reported that Fluvastatin suppresses these cytokines and demonstrates an anti-inflammatory effect. Can the authors provide an explanation for this discrepancy?

4.     The boilerplate words in the Introduction section (lines 40-47) are confusing. These should be deleted. Same for the title. Please double check and remove those.

Comments on the Quality of English Language

The manuscript is generally well-written; however, the authors should meticulously review the wording for accuracy. For instance, the standard or boilerplate phrases in the Introduction section (lines 40-47) are confusing and should be omitted.

Author Response

Response to the comments of referee Number 3:

  1. The scale bars in figures 4 & 5 are unclear. These scale bars need to be carefully reviewed and aligned accurately. For instance, the scale bar in Fig 5C-4 doesn’t seem like 5um.

As suggested by the referee, scale bars in fig. 5 have been meticulously reviewed. We did not find scale bar errors with the exception of figure Fig 5C-4 that has been corrected. Besides, scale bars have been aligned and highlighted. We apologize for these errors.

  1. In Figure 5, when both the autofluorescence and specific signals were displayed in the green channel, it was not convincing enough to distinguish between the auto-fluorescent and specific signals as the authors described. The authors should use another channel to detect caspase-1. Additionally, instead of explaining in the figure legends, proper labeling of staining and channels directly on the pictures is necessary.

We really appreciate this comment. Unfortunately, there is no specific caspase-1 inhibitor available linked to any fluorophore with UV emission, which would be the only fluorescence that would not interfere with bacterial autofluorescence. For better visualization of caspase-1 activity versus bacteria, we have included the following supplementary figure in the new version of the manuscript.

Figure S3. Caspase-1 activation in fluvastatin treated macrophages exposed to Mtb. A) The photomicrographs show the image presented in Figure 5D1 magnified and with the fluorescence of the green and red channels merged or not. B) This image shows the presence of several treated macrophages polarized by the bacteria to perform different functions in the same microenvironment. The image with merged fluorescence shows a macrophage that has captured large amounts of bacteria (in yellow), which apparently enters into its nucleus, without showing specific green fluorescence (white arrow head). Other macrophages (white starts) do not contain autofluorescent bacteria or their remnants, nor do they show specific green fluorescence. The remaining macrophages show a variable number of organelles enclosing both numerous fragments of degraded bacteria (red dots) and specific green fluorescence. In A and B, white arrows show some organelles enclosing active caspase-1 that accumulate bacilli remnants forming arcs at their edges.

  1. In this manuscript, the authors observed that Fluvastatin increased the production of inflammatory cytokines (Fig 6A, IL-1β & IL-18). However, several other studies (PLoS One.2011;6(8):e22655; Acta Cardiol. 2010 Jun;65(3):285-9.; Hepatol Res. 2013 Jul;43(7):775-84) have reported that Fluvastatin suppresses these cytokines and demonstrates an anti-inflammatory effect. Can the authors provide an explanation for this discrepancy?

Statins, by inhibiting one of the most important enzymes of the mevalonate pathway, reduce the synthesis of metabolites (isoprenoids) that play important roles in cell functionality. In our experience, when PBMC are treated with concentrations of fluvastatin higher than 5mM macrophages cells viability could be altered (see reference 22 of the manuscript). The conditions of the three studies cited by the referee are significantly different from those in our study.

In the first article indicated by the referee, fluvastatin reduce IL-8, an interleukin not studied in our work. Moreover, in cells from healthy donors this effect occurs at concentrations higher than 30mM.

In the second paper, the authors found that fluvastatin therapy reduces interleukin-18 and interleukin-10 levels in serum of patients with acute coronary syndrome.  In these patients, cholesterol crystals that accumulated in the arterial intima are considered to be the injurious agent that promotes inflammation by activating NLRP3 inflammasomes (DOIhttps://doi.org/10.1007/s11926-012-0313-z). As the main consequence of the action of statins is to reduce cholesterol levels, by reducing the injurious agent the drug would also reduce IL-1β and IL-18 levels in these patients.

In the third paper the study is performed in primary cultures of hepatocytes, exposed or not to IL-1β in combination with fluvastatin at concentrations 100 mM. This concentration is too high and would affect the functionality of hepatocytes. Furthermore, the authors evaluate the induction of inducible nitric oxide synthase which is not related to the NLRP3 inflammasome. Thus, the model is not comparable to our study.

  1. The boilerplate words in the Introduction section (lines 40-47) are confusing. These should be deleted. Same for the title. Please double check and remove those.

Thanks a lot for your kind indications. Lines 40-47 have been deleted as well as boilerplate words in the title

Round 2

Reviewer 2 Report

Comments and Suggestions for Authors

Thank you for your response to my comments.

The quantitation of the microscopy results doesn't necessarily need counting of cells, but could be scoring of the effects you comment on (necrotic cores, cell aggregates, etc) blinded on treatment. If there is a difference in the treatments it should be quantifiable based on the parameters you are describing.

While you have made some changes to the discussion, it is still too long, and should be focused on the results of the paper.

Reviewer 3 Report

Comments and Suggestions for Authors

The authors have addressed my concerns. I recommend publishing the manuscript with some minor language modifications

For example:

In the Abstract section, on line 34: "We conclude that the use of statins enhances macrophage efficacy in controlling Mtb, further supported by adaptive immunity..." needs clarification. It's unclear whether "macrophage efficacy" or "Mtb" is "supported by adaptive immunity".

On Line 632-634, The sentence "Then, these macrophages die by necrosis releasing viable pathogens into an extracellular milieu where, paradoxically, they grow and spread further" is confusing. 

Comments on the Quality of English Language

Some minor language modifications are needed.